# RoundTable: Investigating Group Decision-Making Mechanism in Multi-Agent Collaboration

## Abstract

This study investigates the efficacy of Multi-Agent Systems in eliciting cross-agent communication and enhancing collective intelligence through group decision-making in a decentralized setting. Unlike centralized mechanisms, where a fixed hierarchy governs social choice, decentralized group decision-making allows agents to engage in joint deliberation. Our research focuses on the dynamics of communication and decision-making within various social choice methods. By applying different voting rules in various environments, we find that moderate decision flexibility yields better outcomes. Additionally, exploring the linguistic features of agent-to-agent conversations reveals indicators of effective collaboration, offering insights into communication patterns that facilitate or hinder collaboration. Finally, we propose various methods for determining the optimal stopping point in multi-agent collaborations based on linguistic cues. Our findings contribute to a deeper understanding of how decentralized decision-making and group conversation shape multi-agent collaboration, with implications for the design of more effective MAS environments.

## 1 Introduction

Collaboration is a fundamental aspect of the nature and human society. Whether among humans or animals, working together allows groups to overcome individual limitations and achieve greater collective outcomes. In nature, collaboration often arises as a strategy to boost survival, enhance resource gathering, or increase efficiency in completing tasks (Schmidt & Mech, 1997). Similarly, in human societies, collaboration drives innovation, facilitates problem-solving, and fosters shared understanding, enabling individuals to address complex challenges that would be unmanageable otherwise (De Man & Duysters, 2005; Graesser et al., 2018; Bittner & Leimeister, 2013). This innate tendency to collaborate is evident across various domains, from social communities to technological systems, where multiple entities coordinate their efforts toward a common goal. As we advance in developing intelligent agents, understanding and replicating these collaborative dynamics in artificial systems has become increasingly important, predominantly to cope with the complexity and adaptability seen in real-world interactions.

Agents powered by Large Language Models (LLMs) have demonstrated impressive problem-solving capabilities across a wide range of tasks. However, single-agent systems encounter significant difficulties when tasked with problems that are either too large or complex, often resulting in instability, misalignment with the intended request, and hallucination (Liu et al., 2024; Kuhn et al., 2023; Lyu et al., 2023). To address these limitations, research has increasingly turned toward Multi-Agent Systems (MAS). MAS have shown greater efficacy in harnessing collective intelligence by allowing individual agents to specialize in distinct skills and facilitating effective collaboration among them (Guo et al., 2024).

When agents working together in a MAS, it is natural for them to have varying interpretations and perspectives. While some opinions may align, disagreements are also frequent. This creates an inevitable tension between cooperation and competition, stemming from differences in backgrounds, information access, and individual goals. Therefore, the process of aggregating models' diverse predictions into a final group decision becomes a crucial aspect of dynamic multi-agent collaboration.

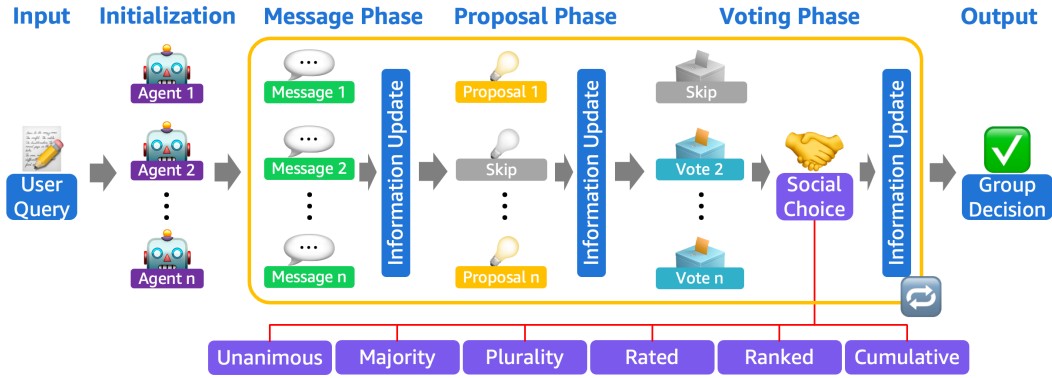

Figure 1: Overview of our multi-agent collaboration platform: RoundTable. It uses a round-based collaboration where agents simultaneously send messages, propose solutions, and vote. Based on a social choice mechanism, RoundTable selects the most preferred proposal for group decisions. Brief introduction of RoundTable is in Section 3.1, details are in Appendix A.1.

Many existing LLM-driven MAS are designed based on centralized group decision-making, which typically involves layered or centralized architectures with a hierarchy. When there is a conflict between agents, it is resolved by a pre-assigned agent or a pre-defined process. These systems are often used for tasks where agent networks follow the waterfall method, leading to a stratified arrangement among agents (Hong et al., 2023; Qian et al., 2023; Dong et al., 2023). However, this hierarchical setup poses several critical challenges: (1) fairness: individual agent messages may not be accurately represented in the final group outcome, leading to potential misrepresentation (Jiang & Lu, 2019); (2) rigidity: the system's fixed structure may lead to over-fitting to specific scenarios (Chen et al., 2023), reducing its adaptability to diverse and dynamic environments; and (3) bias: the agent with final decision-making power may introduce its own biases into the process, potentially distorting the outcomes (Owens et al., 2024). Additionally, centralized MAS cannot perform in environments requiring independent agent decisions, where private or incomplete information hinders a central authority from dictating optimal strategies (Xu et al., 2023).

Decentralized group decision-making can ease these issues by distributing power among agents, where each agent has the ability to participate in the process. This is a common setting in world simulation and embodied environment, where agents need to behave independently because there exists information asymmetry or data boundary between agents (Mandi et al., 2024; Zhang et al., 2023a; Xu et al., 2023; Park et al., 2023), and possibly variability in capabilities that different agents have. With decisions made by multiple agents, the flexible structure of decentralized MAS adapts to various environments, but the lack of a fixed hierarchy demands careful monitoring of collaboration patterns.

Centralized and decentralized group decision-making attempt to mimic the ways in which human societies form collective policies, such as in monarchies and democracies. These decision-making mechanisms, known as social choice, are studied across various fields including economics, mathematics, philosophy, and social science, with the goal of aggregating and synthesizing individual preferences into a unified consensus. However, the impact of social choice methods on LLM-based MAS has yet to be explored.

In this study, we evaluate various social choice methods across different environments to observe and analyze agents' group behaviors and collaboration pattern. This paper aims to provide the following research contributions:

- We investigate how collaborative behavior in decentralized MAS varies across social choice methods, providing insights into how they influence overall cooperation and outcomes.
- We identify key language features in multi-agent conversations as indicators of collaboration, offering a novel approach to analyzing linguistic cues in effective or ineffective interactions.
- We propose various methods for determining the optimal stopping point in multi-agent collaboration, utilizing the linguistic features we identified.

## 2 RELATED WORKS

**Multi Agent Frameworks**   Recent advancements in MAS have led to the development of numerous platforms that facilitate collaboration among multiple agents. AutoGen (Wu et al., 2023) introduces a framework for building LLM-based applications where agents communicate with each other to accomplish tasks. CAMEL (Li et al., 2023a) is designed to enable autonomous collaboration among chat-based language models using role-playing and inception prompting. ChatDev (Qian et al., 2023) offers a chat-driven framework where LLM-powered agents work together to streamline software design, coding, and testing through unified language communication. MetaGPT (Hong et al., 2023) provides a meta-programming approach that encodes Standardized Operating Procedures into prompt sequences, facilitating efficient and error-reducing collaboration among LLM-based agents. AgentVerse (Chen et al., 2023) orchestrates collaborative groups of expert agents, drawing inspiration from human group dynamics to enhance task performance beyond the capabilities of individual agents. These MAS platforms address various aspects of the collaborative process, including agent profiling, recruitment, memory, planning, and communication protocols. However, they do not specifically investigate the group decision-making mechanisms within multi-agent interactions as we do in this paper. Some platforms employ centralized methods, and others only rely on unanimous or majority voting.

**Multi-Agent Collaboration Environments**   Multi-agent collaboration has been used to handle various applications. In robotics, there has been a continuous study regarding multi-agent collaboration in embodied environments (Dudek et al., 1996; Ota, 2006; Cena et al., 2013; Vorotnikov et al., 2018). In these studies, multi-agent collaboration in robotics has been explored through various frameworks, such as task allocation, coordination strategies, and communication protocols, to optimize collective performance and adaptability in dynamic environments. For LLM-based MAS, software development is a popular environment which inherently requires diverse expertise, continuous integration, and coordinated teamwork to build complex, functional systems efficiently (Qian et al., 2023; Hong et al., 2023; Dong et al., 2023). Additionally, LLM's strong context understanding and immense parametric knowledge supported various world simulations with multi-agents. Including simulating society (Park et al., 2023; 2022), economy (Li et al., 2023b; Zhao et al., 2023), gaming (Xu et al., 2023; Wang et al., 2023), and psychology (Aher et al., 2023; Zhang et al., 2023b). These works require agents to interact with others in a group, and investigate their cooperate and competitive behavior.

## 3 MULTI-AGENT COLLABORATION AND GROUP DECISION-MAKING

In this section, we first introduce the formal definition of the setting this study uses for MAS, and propose RoundTable, a turn-based multi-agent collaboration platform for evaluating different decentralized group decision-making.

In a collaboration environment, there is a set of individuals (agents) $i \in \mathbb{I}$, and a set of all possible states of the world $\mathbb{X} = \{x_1, x_2, x_3, ...\}$. Then there is an individual utility function $u_i : \mathbb{X} \to \mathbb{R}$ for each agent that maps a state to a real number, representing preferences of agents. Based on the individual utility function, each agent proposes a state with proposal function $p_i = f_p(u_i, C_i, \mathbb{X}), p_i \in \mathbb{X}$, where $C_i$ is the context, including environment information, background, and collaboration history. Similarly, agent can also vote on proposals with voting function $v_{ij} = f_v(u_i, C_i, \mathbb{S} | p_i \in \mathbb{S} \subseteq \mathbb{X}), i, j \in \mathbb{I}$, where $\mathbb{S}$ represents a candidate list of proposals to vote on. With all proposals and agents' votes, a social choice method (function) $F$ decides one group decision $x^* = F(p_i, v_{ij} | i, j \in \mathbb{I})$. For example, when $F = Majority, p_1 = Apple, p_2 = Banana, p_3 = Carrot, v_{1,j}, v_{2,j} = (Yes, No, No), v_{3,j} = (No, Yes, No)$, then $x^* = p_1 = Apple$.

### 3.1 ROUNDTABLE: MULTI-AGENT COLLABORATION PLATFORM

With the definition above, we propose RoundTable, a multi-agent collaboration platform that can take various group decision-making mechanisms. The overview of RoundTable is shown in Figure 1. Due to the limited space, the details of the platform design and the LLM prompt are shown in Appendix A.1 and A.2. Following is a brief introduction of each phase.

- *Input and Initialization*: RoundTable takes initial query (task), and initializes each agent giving agent specific background and utility function (or goals).
- *Collaboration Round*: RoundTable employs round-based agent collaboration, with each round comprising three phases. Each phase occurs in a simultaneous open conversation, where agents act without order, and information is shared with everyone at the end of each phase. The process ends after $R$ rounds.
    - *Message Phase*: Each agent sends a message to its intended recipients.
    - *Proposal Phase*: Each agent proposes a potential solution, with an option to skip.
    - *Voting Phase*: Each agent votes for the candidate list, with an option to skip. The list consists with each agent's latest proposal and latest accepted proposal by the group. With all votes, social choice method chooses the winner. If ties or disqualifies, the decision is deferred.
- *Output*: After the final round, the latest accepted proposal will be selected as the final output of this system.

## 3.2 Social Choice Methods

To investigate the impact of different social choice methods on group decision-making in multi-agent collaboration, we compare the following 6 mechanisms:

- *Unanimous Voting*(Arrow, 2012): The proposal that receives votes from all agents will be selected.
- *Majority Voting*(Arrow, 2012): The proposal that receives votes from more than half of all agents will be selected.
- *Plurality Voting*(Arrow, 2012): The proposal that receives the most votes will be selected.
- *Rated Voting*(Baujard et al., 2018): Each agent assigns ratings on a 5-point Likert scale to all candidate proposals, with 1 being the lowest and 5 being the highest. The proposal with the highest total score will be selected.
- *Ranked Voting*(Arrow, 2012): Each agent ranks all candidate proposals from the most preferred to the least preferred. Social Choice will assign 1, 1/2, 1/3... points to the 1st, 2nd, 3rd... candidates on each ballot. The proposal with the highest total points will be selected.
- *Cumulative Voting*(Black et al., 1958): For $|\mathbb{I}|$ candidate proposals, each agent is given $|\mathbb{I}|$ points to distribute among the proposals as they see fit. The proposal with the highest total points will be selected.

Unanimous, majority and plurality voting are *one-vote mechanisms*, where agents only choose the best candidate from the list; rated, ranked and cumulative voting are *score-based mechanisms*, which ask a nuanced, gradient preference over candidates.

## 4 Experiments

Using RoundTable, we explore multi-agent behavior patterns in $R = 10$ rounds of collaboration. We evaluate RoundTable in two environments: simulated and complex.

### 4.1 Simulated Environment - Exchange Economy

**Introduction** We use an exchange economy for the simulated environment because its advantages make it well-suited for evaluating a decentralized MAS(Varian, 1992). First, it is a plus-sum game where agents seek equilibrium, meaning collaboration opportunities exist if allocation is not yet optimal. Second, multiple equilibria in the market allow dynamic collaboration. Third, while equilibrium doesn't guarantee maximum utility, it must lie within one of the equilibria, helping to assess whether conflicts between agents hinder the group's progress toward the ultimate goal.

Here, $K$ goods and $K$ agents participate in a market, with each good having a quantity of 100. Each agent has a Cobb-Douglas utility function $u_i = \prod a_k^{\theta_k}, \sum \theta_k = 1$, where $a_k$ represents the amount

of good $k$ for agent $i$, and $\theta_k$ reflects the relative preference for each good (Cobb & Douglas, 1928). Agents aim to maximize their individual utility by finding an optimal allocation of goods. From a group perspective, the hidden goal is to maximize the total utility $U = \sum u_i$, which is not revealed to agents. For more on the exchange economy environment, see Appendix C.2.

To reflect common MAS collaboration patterns, we use an asymmetric utility function setup with $K = 3$ agents, where each agent prefers different goods, mimicking real-world scenarios where agents specialize in different areas. Each agent's utility function is $u_i = a_i^{0.8} \prod_{k \neq i} a_k^{\tilde{\theta}}$, where $\tilde{\theta} = \frac{1-0.8}{|\mathbb{I}|}$. Other types of utility function sets are shared in C.3. We report performance as the average of 100 simulations.

**Metrics**    We use various metrics to analyze multi-agent collaboration. For quality, we report the group total utility $U = \frac{\sum u_i}{U_{max}}$, and $U_{max}$ is the largest possible utility achievable in the environment. Efficiency is measured by the area under the curve (AUC), $AUC@n = \sum_{r=1}^{n} \frac{U_r}{U_{max}}$, where $r$ is round. Fairness is assessed by the Min/Max ratio, which compares the smallest individual utility to the largest at round 10: $\frac{u_{min}}{u_{max}}$. Rationality is the ratio of rounds where the proposed allocation's utility exceeds the current one.[1]    Finally, rigidity measures how often an old allocation is kept, $rigidity = \frac{1}{10} \sum_{r=1}^{10} \mathbf{1}_{U_r = U_{r-1}}$.

### 4.2    Complex Environment - Recommendation System

**Introduction**    For the complex environment, we focus on a recommendation system. It reveals unique collaborative behaviors absent in exchange economies. First, strong information asymmetry exists, as no single piece of information can accurately predict ratings. Second, group decision-making fosters diverse problem-solving approaches, with agents both consuming and contributing information in a feedback loop that enhances recommendations.

The task is to predict user ratings for specific items by analyzing historical interactions and identifying data patterns. This task is challenging for a single-agent system due to the overwhelming amount of scattered and limited essential information. In our setting, we divide the background information into several parts and assign each to different agents. The agents must then collaborate by exploring, analyzing, exchanging, and synthesizing their information to reach a comprehensive conclusion.

**Dataset**    We evaluate MAS performance using the MovieLens-100k dataset (Harper & Konstan, 2015). The dataset consists of three main tables: *user*, which contains demographic information of all users; *item*, which includes details of all movies, such as title, release date, and genre; and *data*, which holds 100,000 ratings from 943 users on 1,682 movies. Due to resource constraints, we randomly selected 100 examples from the u1.test split for evaluation. To simplify the data structure, we pre-processed it into three tables: *basicInfo*, *userHistory*, and *movieHistory*. Each table was assigned to a different agent, creating a 3-agent collaboration system. The definition and example of each table can be found in Appendix C.4.

**Metrics**    We use Mean Absolute Error (MAE) and Root Mean Squared Error (RMSE) to represent utility $U_r$, assessing the accuracy of predictions compared to the gold rating at round $r$. To evaluate MAS performance in the recommendation task, we compare it against two baselines: a simple *Always Guess 4*, which predicts the median regardless of input, and a strong State of the Art (*SoTA*) model from Behera & Nain (2023), which employs collaborative filtering with temporal features.

### 4.3    Cross-Agent Conversation Analysis

To understand agent collaboration and analyze their messages, we perform a language analysis focused on four key features. All observations are collected in 3 agents, gpt-4o-mini setting.

- *Message Length* is a basic yet significant metric, reflecting the amount of information an agent conveys. We measure it using word count, as longer messages generally enrich the conversation.

---

[1]This differs from compromise, where proposing lower utility than the previous proposal is natural, but a lower utility than the current allocation is irrational.

- *Message Complexity* assesses how difficult a message is to understand, indicating the depth of the conversation. We use the Flesch-Kincaid grade level to calculate complexity, with higher scores representing more intricate messages (Klare, 1974).

- *Information Difference* measures how much new information is introduced. Effective collaboration should consistently bring new insights, while low information gain signals a stalled conversation. It is calculated as the average cosine distance between messages in the current round and the center embedding of the previous round.

- *Dialogue Acts* are communicative functions that capture actions, intents, and behaviors within messages as discrete states. We design a set of dialogue acts for multi-agent collaboration: *Inform, Request, Confirm, Summarize, Evaluate, Propose, Compromise, Defend, Accept, Decline*, and use LLM-labeling to perform multi-label classification on a target message based on the previous round. Details of the dialogue act labels and prompt can be found in Appendix C.5.

### 4.4 DIALOGUE ACT TRANSITION GRAPH

We constructed transition graphs to visualize the dynamics of dialogue act transitions across rounds, highlighting the most frequent transitions. Each node represents a dialogue act, and directed edges indicate the probability of transitions between acts from one round to the next. The figure displays only the most probable edge per node, excluding self-loops. The detailed definition of nodes and edges is in Appendix C.6.

### 4.5 EARLY STOPPING IN MAS

In the previous experiments, we enforced 10 rounds of collaboration to compare agent behaviors and identify patterns. However, effective early stopping can prevent redundancy, avoid stagnation, and optimize decision-making by reducing unnecessary iterations. In this experiment, we evaluate the following early stopping methods:

- *@10* is our baseline, the final performance after 10 rounds without early stopping.
- *First Agreement* stops collaboration whenever a proposal passes the social choice criterion.
- *Consecutive Agreements* is when no one make additional proposal after a proposal has been accepted in the previous round.
- *Validation Checkpoint* is the average number of rounds that produced the best outcomes in the train set, used as an early stopping criterion for all test sets.
- *Information Difference* is the average embedding distance captured at the round that produced the best outcomes in the train set, and stops collaborations in the test set when the distance became lower than the threshold. This idea is supported by the observation in Section 5.3.1.
- *Dialogue Act* method utilizes pairs of dialogue acts, linking one from the previous round to another from the current round. We perform ordinary least squares (OLS) regression on all such pairs in relation to the performance. The regression coefficients indicate the most impactful pairs that act as stop signals. Further details of the algorithm are provided in Appendix C.7.

We use 5-fold cross validation to compare the methods with *Oracle*: the oracle performance for early stopping, which reflects the best outcomes from each test simulation.

## 5 RESULTS

### 5.1 EXCHANGE ECONOMY

The experiment results show distinct performance differences across social choice methods in Table 1 and Figure 2(a). Score-based mechanisms achieve higher performance early in the collaboration process compared to one-vote mechanisms, indicating that nuanced evaluations better aggregate agent preferences. One-vote mechanisms struggle with lower and less stable early performance due to increased disagreement, which impairs decision-making. Despite differing early trajectories, all social choice methods reach high-quality outcomes by the end. In most cases, performance peaks in the middle rounds, suggesting that early stopping could be advantageous. Rationality scores

Table 1: Exchange economics environment results across different social choice methods. The reported numbers are an average of all simulations, and the numbers in parentheses are standard errors. The smallest and largest value in a category is colored in blue and red. We see that one-vote mechanisms show lower and less stable early performance, due to the higher probability of disagreement.

| | GROUP TOTAL UTILITY@3 | GROUP TOTAL UTILITY@5 | GROUP TOTAL UTILITY@10 | AUC@3 | AUC@5 | AUC@10 | RATIONALITY | MIN/MAX | RIGIDITY |
|---|---|---|---|---|---|---|---|---|---|
| | | | | Model: gpt-4o-mini, # Agent: K = 3 | | | | | |
| Unanimous | **33.94 (3.96)** | **43.92 (3.96)** | 48.48 (3.86) | **21.77 (2.86)** | **30.17 (3.05)** | **38.65 (3.32)** | **35.00 (1.73)** | **87.96 (1.85)** | **93.00 (0.61)** |
| Majority | 79.88 (0.80) | 81.33 (0.72) | 79.61 (0.88) | 64.08 (1.81) | 70.91 (1.23) | 75.67 (0.84) | 23.80 (0.94) | 64.00 (3.07) | 71.30 (1.32) |
| Plurality | 78.70 (1.32) | 79.42 (1.12) | 76.95 (1.24) | 64.17 (1.85) | 70.34 (1.37) | 74.26 (1.11) | 26.50 (1.10) | **58.11 (3.29)** | 69.10 (1.43) |
| Rated | 80.83 (0.49) | 80.85 (0.97) | **79.91 (0.83)** | 74.02 (0.92) | 76.83 (0.68) | 78.63 (0.59) | 19.80 (0.89) | 67.02 (2.97) | 66.70 (1.56) |
| Ranked | **80.92 (0.62)** | **81.41 (0.73)** | 78.40 (1.46) | **77.31 (1.00)** | **78.89 (0.74)** | **79.41 (0.68)** | **19.03 (0.85)** | 65.10 (3.05) | **61.40 (1.93)** |
| Cumulative | 79.05 (1.11) | 81.10 (0.89) | 78.45 (1.15) | 68.60 (1.62) | 73.63 (1.11) | 76.48 (0.86) | 23.13 (1.05) | 59.24 (3.36) | 65.50 (1.60) |

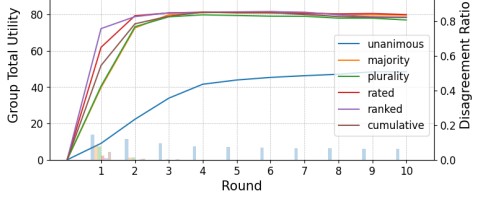

(a) Social choice comparison, using 3 agent, asymmetric, *gpt-4o-mini* setting.

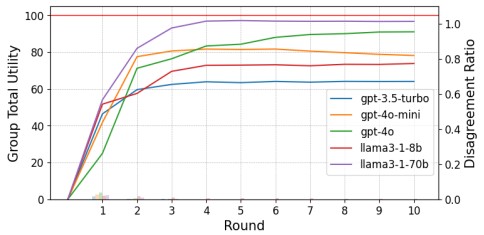

(b) LLM comparison, using 3 agent, asymmetric, majority voting setting.

Figure 2: Exchange economy environment results in group total utility. Line plots (left y-axis) show the group total utility achieved over rounds; bar plots (right y-axis) represent the ratio of cases where participants yet to reach an agreement until a certain round. The red horizontal line indicates the maximum achievable group total utility. We observe that decentralized collaboration performs well across various social choices and LLMs.

show that agents often make suboptimal choices, but decentralized MAS still performs well overall. Score-based mechanisms allow more dynamic decision updates, while one-vote mechanisms exhibit greater rigidity, particularly with unanimous voting.

When comparing one-vote mechanisms, unanimous voting performs the worst due to its inflexibility and need for total agreement, while majority voting slightly outperforms plurality. Additionally, fairness gradually decreases from unanimous to plurality voting. These observations suggest that, compared to the rigid unanimous method or the looser rules of plurality voting, a moderate level of decision flexibility leads to better outcomes, albeit with some sacrifice in fairness across agents.

For the results of ablation studies, please refer to Appendix D.1.

## 5.2 RECOMMENDATION SYSTEM

We illustrate the performance of recommendation system in Figure 3(a), 3(b) and Table 7. The results shows that multi-agent collaboration achieves moderate performance with distributed information, with the decentralized MAS performing comparably to SoTA approaches. While the SoTA method was purely machine-learning-driven, the MAS approach showed reasonable outcomes, indicating that collaboration between agents can explore and produce meaningful signals that help to make decision. Multi-agent collaboration with access to SoTA models may lead to better performance, but we leave this experiment for future work. Similar to the exchange economy, multiple results in the recommendation system follow a V-shaped curve. This suggests that while agent collaboration was efficient early on, it declined in later rounds, highlighting diminishing returns from repeated interactions. The trend underscores the need to balance sustained collaboration within MAS over time.

When comparing one-vote mechanisms to score-based mechanisms, we found that one-vote approaches generally worsened or plateaued after several rounds of collaboration. In contrast, score-

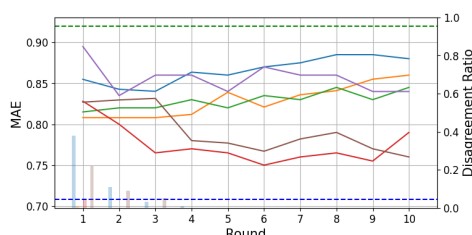 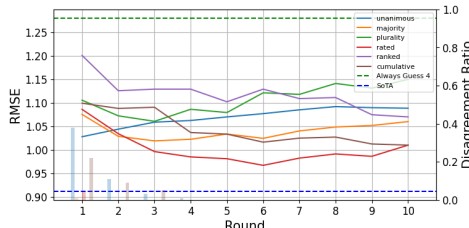

(a) Comparison between social choices in MAE.  (b) Comparison between social choices in RMSE.

Figure 3: Comparison between social choices in recommendation system environment. Line plots (left y-axis) show the comparison of MAE/RMSE achieved over rounds; bar plots (right y-axis) represent the ratio of cases where participants yet to reach an agreement until a certain round.

based mechanisms showed consistent improvement over time, suggesting that score-based mechanisms are better suited for scenarios requiring continuous improvement, allowing for more nuanced input and refinement over multiple rounds.

## 5.3 CROSS-AGENT CONVERSATION ANALYSIS

### 5.3.1 BASIC STATISTICS

Table 8 shows the average values of message length, complexity, and information difference per round. Message length increases over time in both environments, with shorter messages in early rounds and longer ones later. The recommendation system generally has longer messages due to its need for detailed explanations. Message complexity in the exchange economy starts low, peaks early, dips, and then gradually increases toward the final round, while the recommendation system shows consistently higher and increasing complexity throughout. Information difference steadily decreases in both environments, indicating a convergence of topics and less novel information as the discussions progress.

The dialogue act annotation results are presented in Table 9 in Appendix D.4. The dialogue act annotation results in both environments show that *Inform* and *Request* acts dominate, indicating frequent information sharing and input requests critical for task progression. The *Confirm* act rises sharply after round 3 in the recommendation system and more gradually in the exchange economy, showing validation is more prominent in recommendation tasks. *Summarize* acts increase slightly in later rounds, consolidating information, while *Evaluate* acts remain consistently high, reflecting ongoing assessment. *Propose* acts surge early in the exchange economy and peak later in the recommendation system, suggesting early proposals are vital in negotiations. Acts like *Compromise*, *Defend*, *Accept*, and *Decline* are rare, reflecting minimal adversarial behavior, while *Others* remain low, indicating the high quality of dialogue act definitions.

### 5.3.2 DIALOGUE ACT TRANSITION GRAPH

The most probable transition graphs in both environments are shown in Figure 4(a) and 4(b). The breakdown of social choices are displayed in Figure 5 and 6. The transitions reveal a structured progression from requesting information to proposing and evaluating solutions, followed by resolution or negotiation. In the exchange economy, collaboration centers around a loop between *Request* and *Propose*, reflecting the cooperative decision-making process. Key transitions include *Inform* to *Request*, *Confirm* to *Request*, and *Compromise* to *Propose*, showing how agents share knowledge, confirm information, and adopt ideas. *Accept* transitions to *End*, signaling negotiation closure. In contrast, the recommendation system centers on *Request*, indicating a more cooperative, exploratory process with incomplete information.

## 5.4 EARLY STOPPING IN MAS

We show comparison of early stopping methods in recommendation system in Table 2, full results in Table 10, and their basic statistics are in Table 11 in Appendix D.5. To begin, V-shaped performance

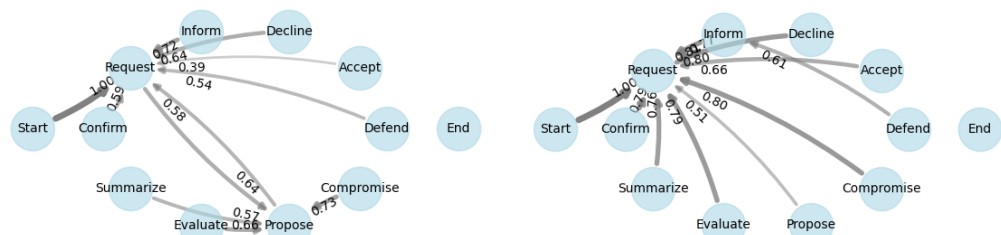

(a) Dialogue act transition graph for exchange economy.

(b) Dialogue act transition graph for recommendation system.

Figure 4: Dialogue act transition graph for exchange economy and recommendation system environments. Only the most probable outgoing edge is presented. Self-loops are excluded.

Table 2: Comparison between early stopping methods in recommendation system environment. The performance is compared with *Oracle* and baseline *@10* in MAE. (↓) indicates better performance with higher and lower values, respectively. Experiments are based on the results of 3 agents, gpt-4o-mini setting. Results are based on 5-fold cross validation. We observe that language-based methods performed well overall.

| EARLY STOPPING METHOD | UNANIMOUS | MAJORITY | PLURALIRY | RATED | RANKED | CUMULATIVE |
|---|---|---|---|---|---|---|
| | RECOMMENDATION SYSTEM, in MAE(↓) | | | | | |
| @10 (Baseline) | 0.88 | 0.86 | 0.84 | 0.79 | 0.84 | **0.76** |
| First Agreement | 0.82 | 0.80 | 0.81 | 0.82 | 0.89 | 0.82 |
| Consecutive Agreements | 0.84 | 0.81 | 0.84 | 0.77 | 0.82 | 0.77 |
| Validation Checkpoint | **0.81** | 0.81 | 0.82 | 0.80 | 0.83 | 0.82 |
| Information Difference | 0.86 | **0.78** | 0.85 | 0.79 | **0.78** | 0.81 |
| Dialogue Act | 0.84 | 0.82 | **0.79** | **0.76** | 0.85 | 0.82 |
| Oracle | 0.73 | 0.63 | 0.59 | 0.62 | 0.69 | 0.67 |

trends were observed across most cases, where the best outcome is observed before reaching the final checkpoint (@10), indicating that intermediate stopping leads to better results than allowing the process to fully complete. This pattern supports the necessity of early stopping in multi-agent collaboration settings, where overextending the interaction can lead to diminishing returns in performance.

Almost all early stopping methods outperformed the baseline (@10), reinforcing the effectiveness of early termination in improving outcomes. Notably, methods leveraging linguistic features, such as Information Difference and Dialogue Act, delivered better performance across different social choices and environments. These findings highlight the advantage of incorporating dialogue-based metrics in deciding when to stop, especially in complex multi-agent environments. Their results across both the exchange economy and recommendation system scenarios further supports the claim that linguistic indicators can be powerful tools for optimizing collaboration outcomes, reducing cognitive load, and improving decision efficiency.

## 6 DISCUSSION

### 6.1 IMPACT OF SOCIAL CHOICE METHODS ON COLLABORATIVE BEHAVIOR

The study demonstrates that social choice methods significantly impact multi-agent collaboration, with score-based mechanisms achieving higher performance and efficiency, especially in early rounds, by allowing agents to express nuanced preferences. In contrast, one-vote mechanisms struggle with rigidity, leading to lower initial performance. The strictness of social choice methods, such as unanimous voting, reduces collaboration performance due to its inability to accommodate dissent. Score-based mechanisms, which support expressing preference gradients, foster dynamic exchanges and improved decision quality, making them more effective in negotiation-heavy tasks.

One-vote mechanisms also have unique advantages. First, the format of voting is simpler than score-based mechanisms, which led to fewer format errors during experiments from LLMs. Furthermore,

unanimous voting showed the highest fairness and non-decreasing outcome over rounds when utility is quantifiable. These strengths suggest that social choice methods in decentralized MAS should be thoughtfully chosen based on the specific task, or the primary metric being prioritized. Mixture of different methods in different stages of collaboration may also bring effectiveness, which we leave for future studies.

## 6.2 LINGUISTIC INDICATORS OF COLLABORATION

Linguistic analysis of agent conversations reveals several key indicators of effective collaboration. The length and complexity of messages increase progressively across rounds, suggesting that deeper engagement and richer exchanges occur as the collaboration advanced. This is particularly evident in the recommendation system environment, where tasks require greater information exchange and negotiation. Complex tasks inherently demand more intricate language, as agents need to convey not just factual information but also interpret, analyze, and synthesize inputs from others.

Dialogue acts provide another layer of insight into collaborative behavior. *Inform*, *Propose* and *Request* acts dominate conversations, highlighting that agents were frequently sharing new information and seeking clarifications, which are essential for progressing towards a solution. *Confirm* act became more prevalent in later stages, particularly in environments where validation was crucial. Interestingly, *Propose* and *Evaluate* acts were more frequent in negotiation-oriented tasks, like exchange economies, where agents needed to offer and assess potential solutions regularly. This linguistic pattern underscores that successful collaboration hinges on the flow of information, clarification, and ongoing evaluation of proposals. These indicators align with patterns seen in human collaboration, where information exchange and continuous feedback loops drive cooperative success.

## 6.3 EARLY STOPPING IN MULTI-AGENT COLLABORATION

Early stopping is essential in multi-agent collaboration because it prevents diminishing returns and inefficiency in decision-making. Our experiments shows that while the initial rounds of collaboration led to significant improvements in group utility and decision quality, continuing beyond a certain point often results in stagnant or even declining performance. This highlights the need for mechanisms that can intelligently terminate the collaboration process once optimal solutions have been reached, avoiding unnecessary iterations.

Key insights into effective early stopping methods emerge from analyzing conversational patterns. Information difference suggests that as the rounds progress, the novelty of information decreases, with fewer new ideas being introduced. Transition graphs of dialogue acts reveal common collaborative patterns. These signals lead to linguistic feature-based stopping methods, which halt collaboration when no meaningful new information is detected, or when dialogue act patterns indicate the need for termination. Our findings demonstrate that implementing early stopping methods based on linguistic cues can enhance the efficiency and effectiveness of decentralized multi-agent collaboration by preventing unnecessary prolonged discussions.

## 7 CONCLUSION

This paper investigates the dynamics of cross-agent communication and decentralized decision-making in MAS, exploring how agent conversation and social choice methods impact collaboration and collective intelligence. We show that only the correct selection of a social choice method for a given environment can promote the success of a MAS, linguistic features of agent conversations serve as indicators of effective collaboration, and these cues provide valuable information for the effective termination of MAS.

LLM generation and prediction are unstable, with single agents often making quasi-random actions based on observations. Decentralized MAS improve performance by guiding agents through conversations to make more targeted proposals and enabling group decision-making to select the best option. Our study provides insights into how group communication and decentralized decision-making enhance multi-agent collaboration, offering important implications for designing more efficient MAS environments.

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

## A  DETAILS OF ROUNDTABLE

### A.1  ROUNDTABLE PLATFORM DESIGN

In this section, we explain the details of the design in RoundTable by breaking it down to phases.

**Input and Initialization** defines each agent for collaboration. Each agent will be given a specific $C_i$ and $u_i$, which include the same input user query, information of RoundTable, and the social choice method $F$. However, they may have different background, information access, or utility function. While $C_i$ and $u_i$ can be defined automatically by agent recruiting or profiling (Chen et al., 2023), we manually define them for simplicity.

**Message Phase** is the initial stage of the iterative collaboration process, where all agents participate in a simultaneous, open chat conversation. During this phase, agents can freely choose recipients based on their context to send messages. They can respond to others' questions, share insights, or ask their own questions. There are no restrictions on whom to communicate with or what to say. Once generated, all messages are updated to $C_i$ for everyone, regardless of the original sender or intended recipients. This process guarantees that all messages are sent simultaneously, makes it has no specific order of communication. Since only one message can be sent per iteration, back-and-forth conversations can take place across multiple rounds.

**Proposal Phase** allows agents to present a proposal $p_i$, offering a potential solution or response to the user query from their perspective, aiming to fulfil their own goal or maximize the utility. Before generating a proposal, agents will be asked to do a step by step reasoning, which is by default invisible to other agents. Reasoning is designed to give agents a chance to summarize and analyze information in $C_i$. The most recent proposal from each agent will be considered as a candidate for voting in the next phase. Like the Message Phase, proposals are made simultaneously and will be updated for $C_i$ once everyone has generated a response. Agents also have the option to skip submitting a new proposal; in such cases, their previous proposal will automatically be included among the candidates.

**Voting Phase** collects one latest proposal $p_i$ from each agent, additionally include the latest intermediate group decision $x^*$ from previous rounds to form a candidate proposal list. Agents in this round are asked to express preferences toward candidates. The preferences can be in various formats, including voting, rating and ranking, depends on what types of social choice method $F$ is used. Similar with the proposal phase, agents have option to not vote (give up). With all the collected proposals and preferences, social choice method selects one proposal as the intermediate group decision. Finally, voting details and the result are updated to all agents' context $C_i$.

**Output** The iteration will run for 10 rounds. After the 10th round, the latest intermediate group decision will be selected as the final output.

## A.2 ROUNDTABLE PROMPT DESIGN

We present the prompt design for RoundTable in this section. Italicized words represent inputs or variables within the prompt.

---

**Initialization Prompt**

# Agent Initialization
You are *my_name*, an agent in a recurring collaboration environment designed to address and solve complex problems.

# Task Description
*task_description*

# Collaboration Rules
You start with nothing decided. The intermediate result will be decided by the social choice function at the end of each round.
In each round, the collaboration runs in 3 phases with the following order:
1. Message Phase: At the beginning of each round, you can send one message to a shared channel for either Talking to one or more agents. All agents will send messages simultaneously. You will be able to see all messages from all agents after the end of the message phase.
2. Proposal Phase: After the end of the message phase, you will have the opportunity to propose potential solution. If you don't propose in this phase, your latest proposal will be used for voting.
3. Voting Phase: At the end of the round, all agents' latest proposal will be voted. When agents didn't propose in this round, their latest proposal will be used for voting. All votes will be processed with the social choice function: *name_of_social_choice*, where *explanation_of_social_choice*. If the social choice function selects a proposal, the intermediate result will be updated accordingly. After each round, each agent will be able to see the result of the vote from the previous round and the conversation history from all rounds.

The collaboration will run for *max_rounds* rounds. After the last round, the latest result will be the final result.

# Your Background
*my_agent_backround*

# Game History
## Latest Candidates at Round *latest_candidates_round*:
*latest_candidates*

## Latest Voting Result at Round *vote_history_length*:
*latest_vote_history*

## Latest Approved Proposal:
Proposal *latest_approved_proposal_id* from Round *latest_approved_proposal_round*.

Proposal *latest_approved_proposal_id* Detail:
*latest_approved_proposal_detail*

## Conversation History until Round *conversation_history_length*:
*conversation_history*

---

**Message Phase Prompt**

You are *my_name*, currently in the message phase of round *round_num*. In this phase, you can:
1. Answer questions posed by others.
2. Share your findings or insights.
3. Ask questions to further the discussion.
You may engage in multiple activities using multiple sentences.

Please type your message in the following JSON format: {"target": <list of agent names>, "message": <str, your message>}
Don't generate anything except the JSON format.

**Proposal Phase Prompt**

You are *my_name*, currently in the proposal phase of round *round_num*. You have an opportunity to make a proposal of the potential solution. Whether or not you submit a new proposal, your latest proposal will be considered as a candidate proposal for the voting phase.

You have two options:
1. Make a proposal:
- You can propose a potential solution by the provided format.
2. Do not make a proposal:
- If you do not want to propose a solution, you can return None as your proposal.

Please type your proposal in the following JSON format: {"reason_for_decision": <your step by step reasoning for your decision>, "proposal": *proposal_format_text*, or None}
Don't generate anything except the JSON format.

**Voting Phase Prompt (for Vote-Based Mechanisms)**

You are *my_name*, at the voting phase at round *round_num*.
In this phase, all agents' latest proposal will be voted by *name_of_social_choice*, where *explanation_of_social_choice*. If the social choice function selects a proposal, the intermediate result will be updated accordingly.

You have two actions to choose: vote or not vote.
1. For vote:
- You can only vote for one of the proposals from the candidate list.
2. For not vote:
- You should vote None.
- If you do not want to vote for any of the proposals, you can vote None.
- If there is no proposal, you vote None.

The same proposal proposed by multiple agents will be merged as one proposal.
If there are multiple proposals passed, none of the proposals will be selected.
If no proposals are passed, the current intermediate result will be kept.

The current candidate proposals are as follows:
*proposal_list*

What is your vote? Please answer in the following JSON format: {"reason_for_decision": <your step by step reasoning for your decision>, "decision": <id of the proposal from the candidates you want to vote, or None>}
Don't generate anything except the JSON format.

## B   DETAILS OF SOCIAL CHOICE METHODS

We present the definition and the prompt we use to define different social choice methods.

> **Unanimous Voting Description**
>
> The proposal that receives votes from all agents will be selected. If no proposal receives votes from all agents, no proposal will be selected.

> **Majority Voting Description**
>
> The proposal that receives votes from more than half of all agents will be selected. If no proposal meets this condition, none will be selected.

> **Plurality Voting Description**
>
> The proposal that receives the most votes will be selected.

> **Rated Voting Description**
>
> Each agent assigns ratings on a 5-point Likert scale to all candidate proposals, with 1 being the lowest and 5 being the highest. The proposal with the highest total score will be selected.

> **Ranked Voting Description**
>
> Each agent ranks all candidate proposals from the most preferred to the least preferred. Social Choice will assign 1, 1/2, 1/3... points to the 1st, 2nd, 3rd... candidates on each ballot. The proposal with the highest total points will be selected.

> **Cumulative Voting Description**
>
> For X candidate proposals, each agent is given X points to distribute among the proposals as they see fit. The proposal with the highest total points will be selected.

## C   DETAILS OF EXPERIMENTS

### C.1   SIMPLE AND COMPLEX ENVIRONMENTS

Simulated environments are simplified, abstract representations of the world. They include economic experiments, such as those based on game theory, and games like chess. The primary advantages of simulated environments are: 1. They allow for the emphasis on specific agent behaviors within a controlled setting by modifying the environmental design; 2. The utility function can be mathematically expressed, and the quality of each proposal can be quantified numerically.

Complex environments are real-world applications that are more intricate and challenging, offering a closer reflection of scenarios where agents must handle numerous variables and unpredictability. These include tasks like collaborative problem-solving, negotiations, and real-time strategy games. While evaluation in these settings can be subjective, they provide deeper insights into how agents adapt, cooperate, and make decisions in dynamic, less controlled conditions, highlighting their resilience and versatility.

### C.2   EXCHANGE ECONOMY

Exchange economy environment has multiple unique advantages. First, it is a plus-sum game. Economically, agents seek an equilibrium until no one can improve their utility without reducing others', implying the possibility of further collaboration exists if the allocation is not at equilibrium. Second, there are multiple possible equilibria exist in the market, allowing dynamic collaboration direction. Third, equilibria are not equal to reaching maximum total utility $U_{max}$, but $U_{max}$ must be lying in

one of the equilibria. This can help us to measure if conflicts between agents harmful for the group to reach an ultimate goal.

The task description we use is as follows:

---

**Exchange Economy Task Description**

You will collaborate with other agents in a recurring exchange market game.
There are *num_of_agents* agents in this market: *list_of_agents*.
There are *num_of_goods* goods in the market: *list_of_goods*. Total quantity of each good is as follows: *total_num_of_goods*.
In this game, you will collaboratively decide how to distribute the goods among the agents. Your goal is to maximize your own utility function.

---

Below is the format of an agent's goal:

---

**Exchange Economy Agent Goal**

Your goal is to maximize your individual utility function by communicating, proposing, and voting with other agents. Your utility function is *util_func*

---

## C.3 EXCHANGE ECONOMY: OTHER UTILITY SETS

We use the asymmetric scenario for the main result due to its similarity to real-world applications, here are other utility sets we examined.

*Symmetric* is the case where all agents prefer the same good, mirroring the scenario where agents collaborate with a common goal. The following utility function is applied to each agent: $u_i = a_1^{0.8} \prod_{k \neq 1} a_k^{\tilde{\theta}}, \tilde{\theta} = \frac{1-0.8}{|I|}$

*Uniform* is the case where all agents indifferently prefers all goods, using the utility function: $u_i = \prod a_k^{\tilde{\theta}}, \tilde{\theta} = \frac{1}{|I|}$

We find these two sets of utility functions are not ideal for evaluating multi-agent collaboration. Since all the agents has the same utility function, the oracle allocation among agents to reach maximized group total utility is very close to even split. However, even split is the most frequent proposal from agents while they have no information for other agents' preferences, making most of the result almost perfect. This phenomenon is observed across all LLMs we've tested.

## C.4 RECOMMENDATION SYSTEM

Recommendation system aims to reveal unique collaborative behaviors that exchange economic does not have. First, there exists strong information asymmetry between agents. Any piece of information is not sufficient to correctly predict the rating. Secondly, the group decision-making in these systems encourages diverse approaches to problem-solving. Agents are not only consumers of information but also contributors, creating a feedback loop where the quality of recommendations improves with increased participation and collaboration. Lastly, the value generated in such a system is not solely based on the final recommendation but also on the process of reaching that recommendation. The interactions, negotiations, and information exchanges that occur along the way contribute to the overall effectiveness and satisfaction of the system.

Below is the description of the environment sent to agents:

| movie_id | movie_title | release_date | genre |
|----------|-------------|--------------|-------|
| 231 | Batman Returns | 19920101 | ['Action', 'Adventure', 'Comedy', 'Crime'] |

| user_id | age | gender | occupation | state |
|---------|-----|--------|------------|-------|
| 7 | 29 | F | artist | NY |

Table 3: Example of *basicInfo* dataset, which includes basic information about the target user and target movie to predict.

---

**Recommendation System Task Description**

You will collaborate with other agents in a movie recommendation game.
In this game, you will collaboratively predict the rating of a target movie (*target_movie_title*) for a target user.
There are 3 agents in this game: BasicInfo Agent, MovieHistory Agent, UserHistory Agent.
1. BasicInfo Agent has access to the basic information of the target movie and target user. It has access to the data with the following schema:
*get_schema(movie_info)*
*get_schema(user_info)*
2. MovieHistory Agent has access to the rating history of the target movie from other people. It has access to the data with the following schema:
*get_schema(movie_rating_history)*
3. UserHistory Agent has access to the rating history of the target user to other movies. It has access to the data with the following schema:
*get_schema(user_rating_history)*

You can't see other agents' information directly, but you can get information from other agents through communication. Your goal is to predict the rating a target user would give to *target_movie_title*. Utilize all available information about both the user and the movie to make the most accurate prediction possible. You only have access to partial information, but you can communicate with other agents to get more information.

---

In the recommendation system enviroment, all agents share the same goal, but has different background dataset. Here is the goal used in our setting:

---

**Recommendation System Agent Goal**

Your goal is to predict the rating the target user would give to the target movie. Utilize all available information about both the user and the movie to make the most accurate prediction possible. You only have access to *data_access*, but you can communicate with other agents to get more information.

# Your Data:
*agent_dataset*

---

To ease the complexity of the data structure, we pre-processed the data into three parts: **basicInfo**, which contains the basic details of the target user and movie; **userHistory**, which includes the target user's ratings and basic information for other movies; and **movieHistory**, which organizes all ratings from other users for the target movie, ranked by a preference similarity score calculated using non-negative matrix factorization on the co-occurrence rating table between users and movies. In our experiment, each part of the dataset is assigned to a different agent, forming a 3-agent collaboration system. Examples of three datasets are in Table 3, 4 and 5.

## C.5    Dialogue Act Labeling

We present the definition of each dialogue act we use in the experiments. The definitions are also used in LLM prompt for automatic annotation.

Conversation Acts (Informational):

| movie_id | movie_title | genre | release_date | rating | rated_date |
|---|---|---|---|---|---|
| 1 | Toy Story | ['Animation', "Children's", 'Comedy'] | 19950101 | 3 | 19980331 |
| 14 | Postino, Il | ['Drama', 'Romance'] | 19940101 | 3 | 19980331 |
| 24 | Rumble in the Bronx | ['Action', 'Adventure', 'Crime'] | 19960223 | 3 | 19980331 |
| 50 | Star Wars | ['Action', 'Adventure', 'Romance', 'Sci-Fi', 'War'] | 19770101 | 4 | 19980331 |
| 109 | Mystery Science Theater 3000: The Movie | ['Comedy', 'Sci-Fi'] | 19960419 | 3 | 19980331 |
| | | ... | | | |

Table 4: Example of *userHistory* dataset, which includes rating history from the target user to other movies. Here only shows 5 rows for spacing.

| user_id | user_pref_similarity | personal_average_score | age | gender | occupation | state | rated_date | rating |
|---|---|---|---|---|---|---|---|---|
| 343 | 0.98 | 3.99 | 43 | M | engineer | GA | 19971009 | 5 |
| 806 | 0.98 | 3.64 | 27 | M | marketing | NY | 19971217 | 3 |
| 773 | 0.98 | 3.28 | 20 | M | student | MN | 19980227 | 2 |
| 805 | 0.98 | 3.35 | 27 | F | other | DC | 19971209 | 3 |
| 447 | 0.97 | 3.57 | 30 | M | administrator | MN | 19971106 | 2 |
| | | | | | ... | | | |

Table 5: Example of *movieHistory* dataset, which includes rating history of the target movie from other users. Here only shows 5 rows for spacing.

- *Inform* - Shares new information that wasn't previously known.

- *Request* - Asks for information that the speaker doesn't have.

- *Confirm* - Asks to verify or validate shared information.

- *Summarize* - Provides a brief overview of the main points.

- *Evaluate* - Gives an opinion or judgment about the information.

Collaboration Acts (Decision-Making):

- *Propose* - Introduce a new solution in the discussion.

- *Compromise* - Offers a balanced solution that incorporates parts of different parties' preferences.

- *Defend* - Maintain support for an idea or solution after consideration or challenge.

- *Accept* - Agrees to or accept an idea or solution.

- *Decline* - Refuses or disagrees with an idea or solution.

## C.6 DIALOGUE ACT TRANSITION GRAPH

Here, we formally define dialogue act transition graph. Two nodes and the directed edge between them consist with a pair of dialogue acts and its transition probability. We denote a pair of dialogue acts as $A \to B$, where $A$ is a dialogue act observed from an agent's message in round $r - 1$, and $B$ is the one from other speakers in round $r$. The transition probability, $p_{A \to B}$, is the probability of the existence of $A \to B$ among all observed $A$s To be more specific, a directed edge between dialogue act $A$ and $B$ is calculated by the following equation:

$$p_{A \to B} = \frac{|A \to B|}{|A|} = \frac{\sum_s \sum_{r=1}^{10} |\{(i, \neg i) | i \in \mathbb{I}, A \in \mathbb{DA}_{i,r-1}, B \in \mathbb{DA}_{\neg i, r}\}|}{\sum_s \sum_{r=1}^{10} |\{i | i \in \mathbb{I}, A \in \mathbb{DA}_{i,r-1}\}|} \quad (1)$$

Where $s$ is the index of simulations, $\mathbb{DA}_{i,r}$ is a set of dialogue acts from the message of the agent $i$ at round $r$, and $\mathbb{DA}_{r=0} = \{Start\}, \mathbb{DA}_{r=11} = \{End\}$.

## C.7 EARLY STOPPING IN MAS

Details of **Dialogue Act** early stopping method is as follows. We first collect all pairs from dialogue acts as independent variable (with one-hot encoding), where one form round $r - 1$ and another from $r$. Then make the group performance at round $r$ to match with dialogue act pairs as dependent variable. Next, we run OLS regression on this dataset, assigning coefficients and p-value to each

Table 6: Ablation studies on exchange economics environment. The reported numbers are an average of all simulations, and the numbers in parentheses are standard errors. The smallest and largest value in a category is colored in blue and red.

| | GROUP TOTAL UTILITY@3 | GROUP TOTAL UTILITY@5 | GROUP TOTAL UTILITY@10 | AUC@3 | AUC@5 | AUC@10 | RATIONALITY | MIN/MAX | RIGIDITY |
|---|---|---|---|---|---|---|---|---|---|
| | | | | Social Choice: Majority, # Agent: K = 3 | | | | | |
| gpt-3.5-turbo | 62.44 (2.24) | 63.40 (0.35) | 64.07 (0.61) | 56.15 (2.81) | 59.15 (1.69) | 61.56 (0.88) | 17.88 (2.11) | 97.58 (0.91) | 47.00 (3.08) |
| gpt-4o-mini | 80.61 (1.46) | 81.43 (1.32) | 78.18 (1.64) | 66.64 (3.07) | 72.59 (2.10) | 76.18 (1.49) | 25.89 (2.25) | 58.41 (6.14) | 70.00 (2.84) |
| gpt-4o | 76.39 (4.26) | 84.29 (1.94) | 91.02 (1.15) | 57.51 (3.82) | 68.03 (2.34) | 78.97 (1.16) | 38.16 (4.19) | 74.27 (3.44) | 73.00 (1.93) |
| llama3-1-8b | 69.55 (5.87) | 72.91 (5.45) | 73.80 (5.39) | 59.60 (6.22) | 64.91 (5.55) | 69.07 (5.08) | 3.78 (0.71) | 77.66 (5.20) | 86.00 (1.83) |
| llama3-1-70b | 93.10 (3.28) | 97.15 (0.56) | 96.69 (0.79) | 76.41 (4.61) | 84.64 (2.78) | 90.69 (1.43) | 35.11 (4.03) | 87.66 (3.28) | 70.67 (2.67) |
| | | | | Social Choice: Majority, Model: gpt-4o-mini | | | | | |
| K = 3 | 79.88 (0.80) | 81.33 (0.72) | 79.61 (0.88) | 64.08 (1.81) | 70.91 (1.23) | 75.67 (0.84) | 23.80 (0.94) | 64.00 (3.07) | 71.30 (1.32) |
| K = 4 | 58.29 (2.39) | 63.10 (2.12) | 64.99 (2.12) | 43.64 (2.04) | 51.05 (1.87) | 57.66 (1.89) | 34.25 (1.39) | 64.54 (3.14) | 80.30 (0.96) |
| K = 5 | 63.67 (2.72) | 72.42 (2.33) | 74.56 (2.18) | 40.23 (1.90) | 52.17 (1.88) | 63.13 (1.94) | 36.26 (1.36) | 57.54 (3.28) | 73.50 (1.36) |

pair. The strength of coefficient shows how each dialogue act pairs relate the performance, positively and negatively, and p-value shows the statistical significance of these relationships.

To find the best early stopping rounds, we do a greedy search on training set for the following hyperparameters:

- *top_da*: the number of dialogue act pairs with the top coefficients to apply in candidates. 1 to 5, inclusive.

- *p-value*: the threshold of p-value for a pair to be considered as candidates. 0.05, 0.1, 0.2 and None.

- *count_per_round*: the threshold of how many occurrences of a candidate should exist in a round to pass. When passed, it gives +1 score for termination. 1 to 3, inclusive.

- *score*: the threshold of how many scores a round needs to be terminated. 1 to value of top_da, inclusive.

We examine all combinations of hyperparameters on the training data, and identifies the best sets for each social choice and environment. Using the best hyperparameters, we evaluate the performance of early stopping on the test set.

## C.8 MODELS

For all experiments, we by default use *gpt-4o-mini-2024-07-18* (OpenAI, 2023) as the main LLM for agents. In ablation study, we use *gpt-3.5-turbo-0125*, *gpt-4o-2024-05-13*, *Llama-3.1-70b-Instruct* and *Llama-3.1-8b-Instruct* for comparison (Dubey et al., 2024). To calculate sentence embeddings we used *paraphrase-MiniLM-L6-v2* (Reimers & Gurevych, 2019). For dialogue act labeling, we use *Llama-3.1-8b-Instruct*. For all LLM inferences, we used *temperature = 0*.

## D DETAILS OF RESULTS

### D.1 ABLATION STUDIES

The result of ablation studies is shown in Table 6. We further compare the performance of multi-agent collaboration with different LLMs in Figure 2(b). For simplicity, we only compare $K = 3$ agent system in majority voting setting with 30 simulations. The results of *gpt-3.5-turbo-0125*, *gpt-4o-2024-05-13*, *Llama-3.1-8b-Instruct* and *Llama-3.1-70b-Instruct* in Table 1 show that multi-agent collaboration works with different LLMs. The stronger (larger) the model, the better performance observed in collaboration, both in end round performance and also efficiency in reaching to an agreement. Notably, the Rationality of the weaker model, *gpt-3.5-turbo-0125* and *Llama-3.1-8b-Instruct*, is only 17.88% and 3.78%, indicates that single agent will largely fail due to the complexity of the task. Nonetheless, the final round performance has reached 64% and 80%, shows the effectiveness of multi-agent collaboration with weak models.

Scaling on MAS is evaluated with different number of agents. From the nature of decentralized collaboration, the more the agents, the harder to reach an agreement. Furthermore, the additional

Table 7: Recommendation system environment results comparison between social choices. The smallest and largest value in a category is colored in blue and red. Experiments are made with gpt-4o-mini.

| | MAE@1 | MAE@2 | MAE@3 | MAE@4 | MAE@5 | MAE@6 | MAE@7 | MAE@8 | MAE@9 | MAE@10 |
|---|---|---|---|---|---|---|---|---|---|---|
| Unanimous | 0.85 | 0.84 | 0.84 | 0.86 | 0.86 | 0.87 | 0.88 | 0.89 | 0.89 | 0.88 |
| Majority | 0.81 | 0.81 | 0.81 | 0.81 | 0.84 | 0.82 | 0.84 | 0.84 | 0.85 | 0.86 |
| Plurality | 0.81 | 0.82 | 0.82 | 0.83 | 0.82 | 0.83 | 0.83 | 0.84 | 0.83 | 0.84 |
| Rated | 0.83 | 0.80 | 0.77 | 0.77 | 0.77 | 0.75 | 0.76 | 0.77 | 0.76 | 0.79 |
| Ranked | 0.90 | 0.83 | 0.86 | 0.86 | 0.84 | 0.87 | 0.86 | 0.86 | 0.84 | 0.84 |
| Cumulative | 0.83 | 0.83 | 0.83 | 0.78 | 0.78 | 0.77 | 0.78 | 0.79 | 0.77 | 0.76 |

| | RMSE@1 | RMSE@2 | RMSE@3 | RMSE@4 | RMSE@5 | RMSE@6 | RMSE@7 | RMSE@8 | RMSE@9 | RMSE@10 |
|---|---|---|---|---|---|---|---|---|---|---|
| Unanimous | 1.03 | 1.04 | 1.06 | 1.06 | 1.07 | 1.08 | 1.09 | 1.09 | 1.09 | 1.09 |
| Majority | 1.08 | 1.03 | 1.02 | 1.02 | 1.03 | 1.02 | 1.04 | 1.05 | 1.05 | 1.06 |
| Plurality | 1.11 | 1.07 | 1.06 | 1.09 | 1.08 | 1.12 | 1.12 | 1.14 | 1.13 | 1.15 |
| Rated | 1.09 | 1.03 | 1.00 | 0.98 | 0.98 | 0.97 | 0.98 | 0.99 | 0.99 | 1.01 |
| Ranked | 1.20 | 1.13 | 1.13 | 1.13 | 1.10 | 1.13 | 1.11 | 1.11 | 1.07 | 1.07 |
| Cumulative | 1.10 | 1.09 | 1.09 | 1.04 | 1.03 | 1.02 | 1.02 | 1.03 | 1.01 | 1.01 |

Table 8: Heatmaps of average message length, message complexity, and information difference in each round. Analysis is made in 3 agents, gpt-4o-mini setting. Color gradient is calculated with green as maximum, red as minimum, and white as median value in each table.

| | EXCHANGE ECONOMY | | | | | | | | | | RECOMMENDATION SYSTEM | | | | | | | | | |
|---|---|---|---|---|---|---|---|---|---|---|---|---|---|---|---|---|---|---|---|---|
| | 1 | 2 | 3 | 4 | 5 | 6 | 7 | 8 | 9 | 10 | 1 | 2 | 3 | 4 | 5 | 6 | 7 | 8 | 9 | 10 |
| | **AVERAGE MESSAGE LENGTH** | | | | | | | | | | | | | | | | | | | |
| Unanimous | 46 | 62 | 83 | 85 | 89 | 94 | 96 | 96 | 97 | 94 | 68 | 85 | 94 | 96 | 100 | 101 | 103 | 104 | 105 | 104 |
| Majority | 43 | 62 | 82 | 84 | 87 | 91 | 91 | 93 | 91 | 91 | 67 | 85 | 93 | 99 | 101 | 103 | 105 | 105 | 106 | 105 |
| Plurality | 42 | 59 | 81 | 82 | 84 | 90 | 89 | 91 | 93 | 90 | 67 | 84 | 94 | 99 | 101 | 104 | 105 | 106 | 107 | 106 |
| Rated | 42 | 60 | 82 | 83 | 89 | 91 | 92 | 94 | 94 | 93 | 67 | 86 | 96 | 100 | 101 | 103 | 104 | 106 | 107 | 105 |
| Ranked | 43 | 61 | 80 | 83 | 86 | 90 | 91 | 90 | 91 | 90 | 67 | 84 | 98 | 99 | 102 | 102 | 103 | 104 | 105 | 104 |
| Cumulative | 42 | 61 | 81 | 83 | 86 | 91 | 91 | 92 | 93 | 91 | 68 | 85 | 95 | 98 | 100 | 102 | 103 | 104 | 104 | 103 |
| | **AVERAGE MESSAGE COMPLEXITY** | | | | | | | | | | | | | | | | | | | |
| Unanimous | 7.1 | 10.2 | 9.3 | 9.0 | 9.0 | 9.1 | 9.2 | 9.5 | 9.5 | 9.6 | 10.4 | 10.7 | 11.5 | 11.7 | 11.9 | 12.0 | 12.0 | 12.1 | 12.3 | 12.2 |
| Majority | 7.6 | 10.4 | 9.5 | 9.4 | 9.2 | 9.2 | 9.4 | 9.5 | 9.6 | 9.7 | 10.3 | 10.7 | 11.5 | 11.7 | 11.9 | 12.0 | 12.1 | 12.1 | 12.1 | 12.1 |
| Plurality | 6.8 | 10.5 | 9.4 | 9.1 | 8.9 | 9.1 | 9.2 | 9.3 | 9.4 | 9.4 | 10.3 | 10.6 | 11.5 | 11.7 | 12.0 | 12.0 | 12.0 | 12.1 | 12.1 | 12.1 |
| Rated | 5.8 | 10.4 | 9.4 | 9.1 | 9.1 | 9.2 | 9.4 | 9.3 | 9.5 | 9.6 | 10.3 | 10.7 | 11.6 | 11.8 | 12.0 | 12.0 | 12.2 | 12.2 | 12.3 | 12.3 |
| Ranked | 6.7 | 10.2 | 9.5 | 9.2 | 9.1 | 9.1 | 9.4 | 9.4 | 9.5 | 9.6 | 10.3 | 10.7 | 11.6 | 11.7 | 11.9 | 12.1 | 12.1 | 12.2 | 12.2 | 12.2 |
| Cumulative | 6.9 | 10.6 | 9.4 | 9.3 | 9.1 | 9.2 | 9.4 | 9.5 | 9.6 | 9.7 | 10.6 | 10.6 | 11.5 | 11.7 | 11.7 | 11.9 | 11.9 | 12.0 | 12.1 | 12.1 |
| | **AVERAGE INFORMATION DIFFERENCE** | | | | | | | | | | | | | | | | | | | |
| Unanimous | | 0.23 | 0.16 | 0.13 | 0.13 | 0.12 | 0.11 | 0.11 | 0.10 | 0.10 | | 0.20 | 0.13 | 0.11 | 0.09 | 0.09 | 0.08 | 0.08 | 0.08 | 0.09 |
| Majority | | 0.26 | 0.17 | 0.13 | 0.12 | 0.12 | 0.11 | 0.10 | 0.10 | 0.11 | | 0.19 | 0.13 | 0.11 | 0.10 | 0.10 | 0.09 | 0.09 | 0.08 | 0.09 |
| Plurality | | 0.28 | 0.16 | 0.13 | 0.14 | 0.12 | 0.11 | 0.11 | 0.10 | 0.11 | | 0.21 | 0.14 | 0.11 | 0.10 | 0.09 | 0.09 | 0.09 | 0.08 | 0.09 |
| Rated | | 0.33 | 0.16 | 0.13 | 0.12 | 0.10 | 0.11 | 0.10 | 0.10 | 0.11 | | 0.20 | 0.13 | 0.11 | 0.10 | 0.10 | 0.09 | 0.09 | 0.08 | 0.09 |
| Ranked | | 0.28 | 0.17 | 0.13 | 0.12 | 0.11 | 0.10 | 0.10 | 0.10 | 0.10 | | 0.20 | 0.13 | 0.11 | 0.09 | 0.09 | 0.09 | 0.09 | 0.08 | 0.09 |
| Cumulative | | 0.30 | 0.18 | 0.13 | 0.12 | 0.11 | 0.11 | 0.11 | 0.10 | 0.10 | | 0.20 | 0.13 | 0.10 | 0.09 | 0.09 | 0.09 | 0.09 | 0.08 | 0.09 |

amount of historical context brought by increasing number of agents can also be a burden to the agent's performance. Comparing $K = 3, 4, 5$ settings in Table 1, the quality and efficiency of collaboration generally drops with the increasing number of agents. Exceptionally, performance in $K = 4$ setting is worse than that of $K = 5$. This is because majority voting is harder to achieve an agreement with, even number of participants. With 4 agents, a proposal needs at least 3 votes to get accepted; 5 agent setting also requires 3 votes, but they can ignore up to 2 agents' preferences or agreements.

## D.2 RECOMMENDATION SYSTEM

We show the detailed result between social choices in recommendation system environment in Table 7.

## D.3 CROSS-AGENT CONVERSATION ANALYSIS

### D.3.1 LINGUISTIC STATISTICS

In this section, we show how linguistic features presented in different environments and social choices. Table 8 is heatmaps for average message length, message complexity, and information difference observed in agent conversation.

Table 9: Heatmaps of ratio of dialogue acts in each round. Color gradient is calculated with green as maximum, and white as minimum value in each table.

**UNANIMOUS**

| | EXCHANGE ECONOMY | | | | | | | | | | RECOMMENDATION SYSTEM | | | | | | | | | |
|---|---|---|---|---|---|---|---|---|---|---|---|---|---|---|---|---|---|---|---|---|
| | 1 | 2 | 3 | 4 | 5 | 6 | 7 | 8 | 9 | 10 | 1 | 2 | 3 | 4 | 5 | 6 | 7 | 8 | 9 | 10 |
| Inform | 1.00 | 0.99 | 0.88 | 0.59 | 0.69 | 0.68 | 0.62 | 0.66 | 0.62 | 0.54 | 1.00 | 0.98 | 0.99 | 1.00 | 1.00 | 1.00 | 0.99 | 1.00 | 1.00 | 1.00 |
| Request | 1.00 | 1.00 | 0.96 | 0.94 | 0.89 | 0.91 | 0.89 | 0.89 | 0.85 | 0.82 | 1.00 | 1.00 | 1.00 | 1.00 | 1.00 | 1.00 | 0.99 | 1.00 | 1.00 | 0.98 |
| Confirm | 0.08 | 0.34 | 0.28 | 0.17 | 0.27 | 0.28 | 0.33 | 0.37 | 0.41 | 0.53 | 0.29 | 0.49 | 0.63 | 0.66 | 0.79 | 0.68 | 0.74 | 0.69 | 0.72 | 0.64 |
| Summarize | 0.16 | 0.02 | 0.15 | 0.17 | 0.16 | 0.19 | 0.13 | 0.13 | 0.18 | 0.32 | 0.07 | 0.11 | 0.08 | 0.16 | 0.16 | 0.18 | 0.16 | 0.16 | 0.19 | 0.24 |
| Evaluate | 0.98 | 0.91 | 0.58 | 0.80 | 0.90 | 0.94 | 0.91 | 0.93 | 0.91 | 0.92 | 0.01 | 0.58 | 0.58 | 0.62 | 0.60 | 0.59 | 0.64 | 0.62 | 0.64 | 0.67 |
| Propose | 0.06 | 0.47 | 0.99 | 0.99 | 0.99 | 1.00 | 0.98 | 0.98 | 0.92 | 0.97 | 0.10 | 0.07 | 0.20 | 0.20 | 0.15 | 0.21 | 0.30 | 0.28 | 0.28 | 0.89 |
| Compromise | 0.00 | 0.09 | 0.43 | 0.13 | 0.24 | 0.39 | 0.42 | 0.37 | 0.39 | 0.40 | 0.01 | 0.00 | 0.00 | 0.00 | 0.01 | 0.00 | 0.00 | 0.00 | 0.00 | 0.00 |
| Defend | 0.00 | 0.00 | 0.04 | 0.00 | 0.02 | 0.03 | 0.02 | 0.01 | 0.03 | 0.08 | 0.00 | 0.00 | 0.00 | 0.01 | 0.01 | 0.02 | 0.02 | 0.01 | 0.03 | 0.05 |
| Accept | 0.00 | 0.00 | 0.01 | 0.00 | 0.00 | 0.02 | 0.06 | 0.08 | 0.12 | 0.27 | 0.00 | 0.00 | 0.00 | 0.01 | 0.00 | 0.02 | 0.01 | 0.01 | 0.04 | 0.02 |
| Decline | 0.00 | 0.00 | 0.01 | 0.00 | 0.00 | 0.01 | 0.01 | 0.00 | 0.02 | 0.03 | 0.00 | 0.00 | 0.00 | 0.00 | 0.00 | 0.00 | 0.00 | 0.01 | 0.01 | 0.00 |
| Others | 0.00 | 0.00 | 0.02 | 0.06 | 0.04 | 0.01 | 0.06 | 0.04 | 0.04 | 0.04 | 0.03 | 0.04 | 0.00 | 0.03 | 0.01 | 0.02 | 0.04 | 0.01 | 0.03 | 0.01 |

**MAJORITY**

| | EXCHANGE ECONOMY | | | | | | | | | | RECOMMENDATION SYSTEM | | | | | | | | | |
|---|---|---|---|---|---|---|---|---|---|---|---|---|---|---|---|---|---|---|---|---|
| | 1 | 2 | 3 | 4 | 5 | 6 | 7 | 8 | 9 | 10 | 1 | 2 | 3 | 4 | 5 | 6 | 7 | 8 | 9 | 10 |
| Inform | 1.00 | 1.00 | 0.96 | 0.72 | 0.72 | 0.66 | 0.65 | 0.77 | 0.66 | 0.64 | 1.00 | 1.00 | 1.00 | 1.00 | 1.00 | 1.00 | 1.00 | 1.00 | 0.99 | 1.00 |
| Request | 1.00 | 1.00 | 1.00 | 0.96 | 0.95 | 0.96 | 0.88 | 0.96 | 0.90 | 0.85 | 1.00 | 1.00 | 1.00 | 1.00 | 1.00 | 1.00 | 1.00 | 0.99 | 1.00 | 0.92 |
| Confirm | 0.16 | 0.25 | 0.34 | 0.31 | 0.44 | 0.36 | 0.41 | 0.41 | 0.53 | 0.52 | 0.25 | 0.48 | 0.56 | 0.65 | 0.71 | 0.74 | 0.74 | 0.71 | 0.81 | 0.65 |
| Summarize | 0.15 | 0.02 | 0.15 | 0.13 | 0.19 | 0.21 | 0.20 | 0.18 | 0.33 | 0.30 | 0.01 | 0.15 | 0.07 | 0.18 | 0.11 | 0.17 | 0.19 | 0.16 | 0.18 | 0.25 |
| Evaluate | 0.93 | 0.91 | 0.67 | 0.78 | 0.86 | 0.88 | 0.88 | 0.86 | 0.89 | 0.87 | 0.05 | 0.52 | 0.49 | 0.58 | 0.52 | 0.61 | 0.54 | 0.61 | 0.58 | 0.56 |
| Propose | 0.37 | 0.41 | 0.97 | 0.99 | 0.95 | 0.97 | 0.99 | 0.96 | 0.93 | 0.90 | 0.05 | 0.08 | 0.29 | 0.17 | 0.22 | 0.15 | 0.25 | 0.21 | 0.26 | 0.89 |
| Compromise | 0.00 | 0.03 | 0.38 | 0.14 | 0.25 | 0.41 | 0.39 | 0.31 | 0.36 | 0.30 | 0.00 | 0.00 | 0.02 | 0.00 | 0.00 | 0.00 | 0.00 | 0.00 | 0.00 | 0.00 |
| Defend | 0.00 | 0.00 | 0.04 | 0.01 | 0.04 | 0.06 | 0.07 | 0.05 | 0.09 | 0.07 | 0.00 | 0.01 | 0.01 | 0.00 | 0.02 | 0.01 | 0.02 | 0.00 | 0.03 | 0.04 |
| Accept | 0.00 | 0.00 | 0.02 | 0.03 | 0.04 | 0.11 | 0.08 | 0.12 | 0.20 | 0.20 | 0.00 | 0.02 | 0.02 | 0.01 | 0.01 | 0.00 | 0.02 | 0.01 | 0.02 | 0.03 |
| Decline | 0.00 | 0.00 | 0.02 | 0.02 | 0.03 | 0.02 | 0.04 | 0.03 | 0.04 | 0.00 | 0.00 | 0.01 | 0.02 | 0.00 | 0.00 | 0.00 | 0.01 | 0.00 | 0.01 | 0.01 |
| Others | 0.00 | 0.01 | 0.01 | 0.08 | 0.08 | 0.08 | 0.08 | 0.08 | 0.07 | 0.06 | 0.02 | 0.00 | 0.03 | 0.03 | 0.01 | 0.02 | 0.02 | 0.01 | 0.01 | 0.01 |

**PLURALITY**

| | EXCHANGE ECONOMY | | | | | | | | | | RECOMMENDATION SYSTEM | | | | | | | | | |
|---|---|---|---|---|---|---|---|---|---|---|---|---|---|---|---|---|---|---|---|---|
| | 1 | 2 | 3 | 4 | 5 | 6 | 7 | 8 | 9 | 10 | 1 | 2 | 3 | 4 | 5 | 6 | 7 | 8 | 9 | 10 |
| Inform | 1.00 | 0.98 | 0.93 | 0.74 | 0.69 | 0.68 | 0.61 | 0.64 | 0.66 | 0.64 | 1.00 | 1.00 | 0.98 | 0.99 | 0.99 | 0.99 | 1.00 | 0.98 | 1.00 | 1.00 |
| Request | 1.00 | 1.00 | 0.95 | 0.97 | 1.00 | 0.93 | 0.91 | 0.90 | 0.96 | 0.85 | 1.00 | 1.00 | 1.00 | 1.00 | 1.00 | 1.00 | 0.99 | 1.00 | 1.00 | 0.93 |
| Confirm | 0.01 | 0.29 | 0.16 | 0.26 | 0.40 | 0.42 | 0.43 | 0.46 | 0.52 | 0.55 | 0.29 | 0.44 | 0.61 | 0.70 | 0.73 | 0.78 | 0.76 | 0.78 | 0.77 | 0.61 |
| Summarize | 0.39 | 0.01 | 0.11 | 0.17 | 0.17 | 0.22 | 0.20 | 0.22 | 0.18 | 0.25 | 0.05 | 0.13 | 0.11 | 0.23 | 0.18 | 0.23 | 0.19 | 0.17 | 0.21 | 0.19 |
| Evaluate | 0.85 | 0.96 | 0.62 | 0.73 | 0.86 | 0.85 | 0.91 | 0.89 | 0.83 | 0.86 | 0.06 | 0.65 | 0.59 | 0.61 | 0.69 | 0.58 | 0.62 | 0.67 | 0.65 | 0.61 |
| Propose | 0.19 | 0.49 | 0.99 | 1.00 | 0.98 | 0.97 | 0.97 | 0.95 | 0.93 | 0.87 | 0.09 | 0.06 | 0.31 | 0.31 | 0.30 | 0.32 | 0.32 | 0.34 | 0.36 | 0.85 |
| Compromise | 0.00 | 0.02 | 0.53 | 0.14 | 0.14 | 0.38 | 0.37 | 0.33 | 0.32 | 0.32 | 0.00 | 0.01 | 0.00 | 0.01 | 0.02 | 0.00 | 0.00 | 0.00 | 0.00 | 0.00 |
| Defend | 0.00 | 0.00 | 0.00 | 0.02 | 0.00 | 0.05 | 0.06 | 0.06 | 0.07 | 0.07 | 0.00 | 0.00 | 0.01 | 0.03 | 0.02 | 0.04 | 0.04 | 0.02 | 0.02 | 0.03 |
| Accept | 0.00 | 0.00 | 0.01 | 0.02 | 0.00 | 0.08 | 0.15 | 0.10 | 0.13 | 0.25 | 0.00 | 0.00 | 0.01 | 0.03 | 0.01 | 0.02 | 0.04 | 0.02 | 0.01 | 0.02 |
| Decline | 0.00 | 0.00 | 0.01 | 0.02 | 0.00 | 0.05 | 0.05 | 0.01 | 0.02 | 0.02 | 0.00 | 0.00 | 0.01 | 0.01 | 0.01 | 0.01 | 0.01 | 0.01 | 0.00 | 0.00 |
| Others | 0.00 | 0.00 | 0.02 | 0.10 | 0.06 | 0.08 | 0.05 | 0.09 | 0.07 | 0.04 | 0.01 | 0.01 | 0.02 | 0.01 | 0.06 | 0.02 | 0.07 | 0.00 | 0.03 | 0.01 |

**RATED**

| | EXCHANGE ECONOMY | | | | | | | | | | RECOMMENDATION SYSTEM | | | | | | | | | |
|---|---|---|---|---|---|---|---|---|---|---|---|---|---|---|---|---|---|---|---|---|
| | 1 | 2 | 3 | 4 | 5 | 6 | 7 | 8 | 9 | 10 | 1 | 2 | 3 | 4 | 5 | 6 | 7 | 8 | 9 | 10 |
| Inform | 1.00 | 0.94 | 0.97 | 0.78 | 0.69 | 0.56 | 0.59 | 0.65 | 0.65 | 0.61 | 1.00 | 1.00 | 0.98 | 0.99 | 0.99 | 1.00 | 1.00 | 1.00 | 1.00 | 0.99 |
| Request | 1.00 | 1.00 | 1.00 | 0.97 | 0.96 | 0.92 | 0.93 | 0.91 | 0.90 | 0.94 | 1.00 | 1.00 | 1.00 | 1.00 | 1.00 | 1.00 | 1.00 | 0.99 | 1.00 | 0.94 |
| Confirm | 0.00 | 0.47 | 0.33 | 0.31 | 0.35 | 0.41 | 0.33 | 0.40 | 0.43 | 0.48 | 0.27 | 0.48 | 0.67 | 0.70 | 0.76 | 0.65 | 0.67 | 0.71 | 0.69 | 0.58 |
| Summarize | 0.16 | 0.02 | 0.17 | 0.13 | 0.17 | 0.19 | 0.23 | 0.23 | 0.20 | 0.25 | 0.09 | 0.13 | 0.14 | 0.16 | 0.17 | 0.22 | 0.27 | 0.21 | 0.18 | 0.25 |
| Evaluate | 0.40 | 1.00 | 0.57 | 0.69 | 0.87 | 0.85 | 0.85 | 0.84 | 0.92 | 0.85 | 0.09 | 0.60 | 0.62 | 0.66 | 0.58 | 0.66 | 0.58 | 0.51 | 0.55 | 0.71 |
| Propose | 0.11 | 0.66 | 0.97 | 0.99 | 0.98 | 0.99 | 1.00 | 0.97 | 0.95 | 0.94 | 0.16 | 0.05 | 0.26 | 0.20 | 0.21 | 0.23 | 0.27 | 0.31 | 0.36 | 0.88 |
| Compromise | 0.00 | 0.03 | 0.48 | 0.12 | 0.24 | 0.40 | 0.32 | 0.34 | 0.38 | 0.32 | 0.00 | 0.00 | 0.01 | 0.01 | 0.01 | 0.00 | 0.00 | 0.00 | 0.01 | 0.00 |
| Defend | 0.00 | 0.00 | 0.03 | 0.01 | 0.02 | 0.03 | 0.07 | 0.03 | 0.07 | 0.04 | 0.00 | 0.00 | 0.00 | 0.01 | 0.01 | 0.00 | 0.00 | 0.01 | 0.00 | 0.02 |
| Accept | 0.00 | 0.00 | 0.03 | 0.02 | 0.05 | 0.02 | 0.07 | 0.10 | 0.11 | 0.22 | 0.00 | 0.00 | 0.00 | 0.00 | 0.00 | 0.00 | 0.01 | 0.02 | 0.00 | 0.01 |
| Decline | 0.00 | 0.00 | 0.02 | 0.01 | 0.01 | 0.00 | 0.01 | 0.03 | 0.03 | 0.00 | 0.00 | 0.00 | 0.00 | 0.00 | 0.00 | 0.00 | 0.01 | 0.00 | 0.00 | 0.00 |
| Others | 0.02 | 0.00 | 0.02 | 0.10 | 0.08 | 0.08 | 0.06 | 0.07 | 0.05 | 0.08 | 0.03 | 0.02 | 0.06 | 0.04 | 0.03 | 0.02 | 0.02 | 0.05 | 0.05 | 0.06 |

**RANKED**

| | EXCHANGE ECONOMY | | | | | | | | | | RECOMMENDATION SYSTEM | | | | | | | | | |
|---|---|---|---|---|---|---|---|---|---|---|---|---|---|---|---|---|---|---|---|---|
| | 1 | 2 | 3 | 4 | 5 | 6 | 7 | 8 | 9 | 10 | 1 | 2 | 3 | 4 | 5 | 6 | 7 | 8 | 9 | 10 |
| Inform | 1.00 | 0.96 | 0.96 | 0.78 | 0.79 | 0.66 | 0.69 | 0.71 | 0.72 | 0.75 | 1.00 | 1.00 | 1.00 | 1.00 | 0.99 | 0.99 | 1.00 | 1.00 | 1.00 | 0.99 |
| Request | 1.00 | 1.00 | 0.99 | 0.98 | 0.96 | 0.95 | 0.95 | 0.96 | 0.91 | 0.89 | 1.00 | 1.00 | 1.00 | 1.00 | 1.00 | 1.00 | 1.00 | 1.00 | 1.00 | 0.98 |
| Confirm | 0.03 | 0.51 | 0.49 | 0.33 | 0.40 | 0.35 | 0.45 | 0.52 | 0.48 | 0.54 | 0.21 | 0.43 | 0.53 | 0.62 | 0.73 | 0.75 | 0.78 | 0.68 | 0.70 | 0.70 |
| Summarize | 0.02 | 0.03 | 0.09 | 0.16 | 0.23 | 0.20 | 0.20 | 0.16 | 0.27 | 0.32 | 0.09 | 0.14 | 0.14 | 0.21 | 0.24 | 0.23 | 0.21 | 0.26 | 0.17 | 0.24 |
| Evaluate | 0.92 | 0.95 | 0.61 | 0.65 | 0.79 | 0.86 | 0.87 | 0.86 | 0.83 | 0.89 | 0.06 | 0.52 | 0.64 | 0.66 | 0.63 | 0.58 | 0.59 | 0.60 | 0.69 | 0.77 |
| Propose | 0.15 | 0.48 | 0.92 | 0.96 | 0.99 | 0.98 | 0.97 | 0.93 | 0.90 | 0.87 | 0.09 | 0.04 | 0.27 | 0.16 | 0.29 | 0.19 | 0.24 | 0.28 | 0.32 | 0.85 |
| Compromise | 0.00 | 0.01 | 0.35 | 0.14 | 0.21 | 0.32 | 0.35 | 0.36 | 0.37 | 0.21 | 0.00 | 0.01 | 0.01 | 0.02 | 0.00 | 0.00 | 0.01 | 0.03 | 0.00 | 0.03 |
| Defend | 0.00 | 0.00 | 0.05 | 0.02 | 0.03 | 0.06 | 0.03 | 0.02 | 0.05 | 0.09 | 0.00 | 0.00 | 0.00 | 0.00 | 0.00 | 0.01 | 0.01 | 0.01 | 0.02 | 0.02 |
| Accept | 0.00 | 0.00 | 0.05 | 0.03 | 0.04 | 0.05 | 0.12 | 0.13 | 0.11 | 0.27 | 0.00 | 0.00 | 0.02 | 0.01 | 0.02 | 0.01 | 0.01 | 0.02 | 0.00 | 0.03 |
| Decline | 0.00 | 0.00 | 0.03 | 0.01 | 0.01 | 0.03 | 0.01 | 0.00 | 0.00 | 0.02 | 0.00 | 0.00 | 0.00 | 0.00 | 0.00 | 0.00 | 0.01 | 0.00 | 0.00 | 0.01 |
| Others | 0.01 | 0.02 | 0.02 | 0.05 | 0.06 | 0.13 | 0.11 | 0.04 | 0.03 | 0.01 | 0.01 | 0.02 | 0.08 | 0.03 | 0.01 | 0.03 | 0.01 | 0.03 | 0.02 | 0.06 |

**CUMULATIVE**

| | EXCHANGE ECONOMY | | | | | | | | | | RECOMMENDATION SYSTEM | | | | | | | | | |
|---|---|---|---|---|---|---|---|---|---|---|---|---|---|---|---|---|---|---|---|---|
| | 1 | 2 | 3 | 4 | 5 | 6 | 7 | 8 | 9 | 10 | 1 | 2 | 3 | 4 | 5 | 6 | 7 | 8 | 9 | 10 |
| Inform | 1.00 | 0.94 | 0.99 | 0.82 | 0.72 | 0.68 | 0.60 | 0.67 | 0.72 | 0.74 | 1.00 | 1.00 | 1.00 | 0.98 | 0.97 | 0.98 | 1.00 | 1.00 | 1.00 | 1.00 |
| Request | 1.00 | 1.00 | 0.98 | 0.97 | 0.96 | 0.95 | 0.94 | 0.92 | 0.95 | 0.86 | 1.00 | 1.00 | 1.00 | 1.00 | 1.00 | 0.99 | 1.00 | 1.00 | 1.00 | 0.96 |
| Confirm | 0.03 | 0.32 | 0.30 | 0.29 | 0.29 | 0.33 | 0.41 | 0.46 | 0.45 | 0.55 | 0.25 | 0.44 | 0.66 | 0.72 | 0.78 | 0.73 | 0.73 | 0.76 | 0.71 | 0.62 |
| Summarize | 0.14 | 0.01 | 0.14 | 0.16 | 0.09 | 0.17 | 0.25 | 0.23 | 0.21 | 0.32 | 0.10 | 0.12 | 0.12 | 0.22 | 0.22 | 0.22 | 0.17 | 0.29 | 0.20 | 0.26 |
| Evaluate | 0.81 | 0.98 | 0.58 | 0.74 | 0.86 | 0.90 | 0.92 | 0.90 | 0.90 | 0.86 | 0.04 | 0.65 | 0.63 | 0.51 | 0.61 | 0.62 | 0.59 | 0.56 | 0.60 | 0.72 |
| Propose | 0.17 | 0.58 | 0.98 | 0.97 | 0.99 | 0.98 | 0.95 | 0.89 | 0.89 | 0.87 | 0.08 | 0.05 | 0.24 | 0.26 | 0.22 | 0.22 | 0.27 | 0.28 | 0.32 | 0.84 |
| Compromise | 0.00 | 0.05 | 0.54 | 0.14 | 0.19 | 0.42 | 0.30 | 0.35 | 0.28 | 0.27 | 0.01 | 0.00 | 0.00 | 0.02 | 0.00 | 0.00 | 0.02 | 0.01 | 0.00 | 0.01 |
| Defend | 0.00 | 0.00 | 0.01 | 0.00 | 0.03 | 0.05 | 0.05 | 0.02 | 0.05 | 0.06 | 0.01 | 0.01 | 0.01 | 0.01 | 0.01 | 0.00 | 0.02 | 0.03 | 0.02 | 0.03 |
| Accept | 0.00 | 0.00 | 0.01 | 0.00 | 0.01 | 0.09 | 0.10 | 0.10 | 0.21 | 0.21 | 0.01 | 0.01 | 0.00 | 0.00 | 0.01 | 0.03 | 0.01 | 0.03 | 0.02 | 0.04 |
| Decline | 0.00 | 0.00 | 0.00 | 0.00 | 0.00 | 0.04 | 0.01 | 0.00 | 0.00 | 0.01 | 0.01 | 0.01 | 0.00 | 0.00 | 0.00 | 0.00 | 0.00 | 0.02 | 0.01 | 0.00 |
| Others | 0.00 | 0.00 | 0.00 | 0.06 | 0.06 | 0.07 | 0.05 | 0.06 | 0.06 | 0.03 | 0.01 | 0.03 | 0.04 | 0.03 | 0.01 | 0.07 | 0.01 | 0.03 | 0.04 | 0.08 |

## D.4 DIALOGUE ACTS

## D.5 EARLY STOPPING

We share full result of early stopping experiment in Table 10, and basic statistics of early stopping methods, including early stopped round, effective ratio, and information difference threshold in Table 11. Effective ratio stands for the chance of a certain rule can be applied in the test set. Threshold is the embedding distance threshold used for early stopping.

Although linguistic feature based methods, *Information Difference* and *Dialogue Act*, outperforms other early stopping methods and baseline in exchange economy environment, the difference is small. This is due to the small room for improvement, identified between the baseline (@10) and *Oracle* performance.

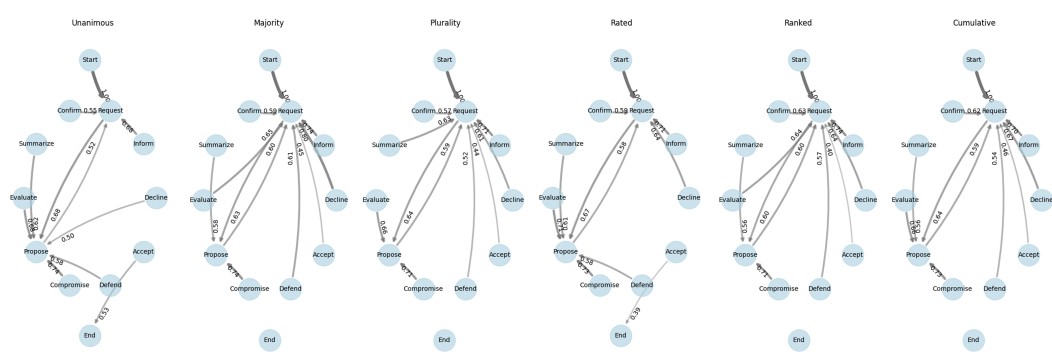

Figure 5: Dialogue act transition graph for different social choices in exchange economics environment.

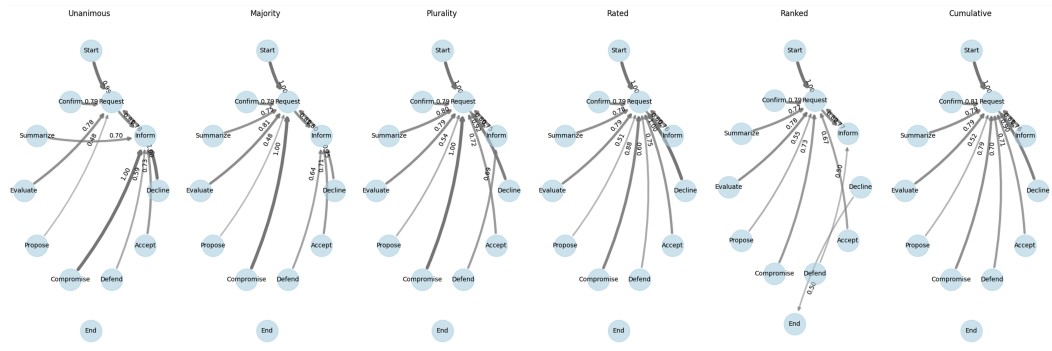

Figure 6: Dialogue act transition graph for different social choices in recommendation system environment.

Table 10: Comparison between early stopping methods in different social choices. The performance is shown with group total utility and MAE. (↑) and (↓) indicates better performance with higher and lower values, respectively. Experiments are based on the results of 3 agents, gpt-4o-mini setting. Results are based on 5-fold cross validation. We observe that language-based methods performed well overall.

| EARLY STOPPING METHOD | UNANIMOUS | MAJORITY | PLURALIRY | RATED | RANKED | CUMULATIVE |
|---|---|---|---|---|---|---|
| **EXCHANGE ECONOMY, in Group Total Utility(↑)** | | | | | | |
| @10 (Baseline) | **0.48** | 0.80 | 0.77 | 0.80 | 0.78 | 0.78 |
| First Agreement | 0.48 | 0.78 | 0.76 | 0.74 | 0.77 | 0.74 |
| Consecutive Agreements | **0.48** | 0.80 | 0.77 | 0.80 | 0.78 | 0.79 |
| Validation Checkpoint | 0.37 | 0.81 | 0.80 | 0.81 | 0.81 | 0.81 |
| Information Difference | 0.39 | **0.81** | **0.80** | 0.81 | 0.81 | **0.82** |
| Dialogue Act | 0.42 | 0.81 | 0.80 | 0.81 | 0.81 | 0.81 |
| Oracle | 0.48 | 0.84 | 0.82 | 0.83 | 0.84 | 0.84 |
| | | | | | | |
| **RECOMMENDATION SYSTEM, in MAE(↓)** | | | | | | |
| @10 (Baseline) | 0.88 | 0.86 | 0.84 | 0.79 | 0.84 | **0.76** |
| First Agreement | 0.82 | 0.80 | 0.81 | 0.82 | 0.89 | 0.82 |
| Consecutive Agreements | 0.84 | 0.81 | 0.84 | 0.77 | 0.82 | 0.77 |
| Validation Checkpoint | **0.81** | 0.81 | 0.82 | 0.80 | 0.83 | 0.82 |
| Information Difference | 0.86 | **0.78** | 0.85 | 0.79 | **0.78** | 0.81 |
| Dialogue Act | 0.84 | 0.82 | **0.79** | **0.76** | 0.85 | 0.82 |
| Oracle | 0.73 | 0.63 | 0.59 | 0.62 | 0.69 | 0.67 |

Table 11: Basic statistics of different early stopping methods.

| STATISTICS | EARLY STOPPING METHOD | UNANIMOUS | MAJORITY | PLURALITY | RATED | RANKED | CUMULATIVE | AVERAGE |
|---|---|---|---|---|---|---|---|---|
| | **EXCHANGE ECONOMY** | | | | | | | |
| | Oracle | 3.50 | 3.61 | 3.84 | 3.81 | 3.49 | 4.11 | 3.73 |
| | First Agreement | 3.00 | 1.57 | 1.54 | 1.15 | 1.07 | 1.33 | 1.61 |
| | Consecutive Agreement | - | 8.00 | 6.50 | 8.00 | 10.00 | 7.00 | 7.90 |
| Early Stopped Round | Validation Checkpoint | 2.60 | 2.80 | 3.00 | 3.00 | 2.60 | 3.00 | 2.83 |
| | Information Difference | 3.21 | 3.32 | 3.28 | 3.09 | 3.31 | 3.66 | 3.31 |
| | Dialogue Act | 6.23 | 7.09 | 8.33 | 7.64 | 5.47 | 7.40 | 7.03 |
| | Ensemble | 10.00 | 4.15 | 4.73 | 7.64 | 5.01 | 3.66 | 5.86 |
| | First Agreement | 0.62 | 1.00 | 1.00 | 1.00 | 1.00 | 1.00 | 0.94 |
| Effective Ratio | Consecutive Agreement | 0.00 | 0.03 | 0.03 | 0.02 | 0.00 | 0.01 | 0.02 |
| | Information Difference | 1.00 | 1.00 | 1.00 | 1.00 | 1.00 | 1.00 | 1.00 |
| | Dialogue Act | 0.62 | 0.34 | 0.07 | 0.19 | 0.63 | 0.31 | 0.36 |
| Threshold | Information Difference | 0.17 | 0.17 | 0.17 | 0.20 | 0.18 | 0.17 | 0.18 |
| | **RECOMMENDATION SYSTEM** | | | | | | | |
| | Oracle | 2.40 | 2.10 | 2.38 | 2.15 | 2.25 | 2.44 | 2.29 |
| | First Agreement | 1.53 | 1.01 | 1.00 | 1.04 | 1.00 | 1.36 | 1.16 |
| | Consecutive Agreement | 3.34 | 3.31 | 3.15 | 3.61 | 3.46 | 3.87 | 3.46 |
| Early Stopped Round | Validation Checkpoint | 1.20 | 1.00 | 1.20 | 1.00 | 1.00 | 1.00 | 1.07 |
| | Information Difference | 3.12 | 3.67 | 4.30 | 3.50 | 3.97 | 3.34 | 3.65 |
| | Dialogue Act | 7.58 | 6.98 | 6.02 | 5.22 | 6.53 | 6.53 | 6.28 |
| | Ensemble | 3.60 | 5.87 | 5.12 | 5.38 | 3.97 | 10.00 | 5.66 |
| | First Agreement | 1.00 | 1.00 | 1.00 | 1.00 | 1.00 | 1.00 | 1.00 |
| Effective Ratio | Consecutive Agreement | 0.82 | 0.92 | 0.91 | 0.77 | 0.62 | 0.78 | 0.80 |
| | Information Difference | 1.00 | 1.00 | 1.00 | 1.00 | 0.99 | 1.00 | 1.00 |
| | Dialogue Act | 0.06 | 0.24 | 0.52 | 0.58 | 0.20 | 0.30 | 0.32 |
| Threshold | Information Difference | 0.15 | 0.13 | 0.12 | 0.14 | 0.12 | 0.14 | 0.13 |

