# OpenReview forum: "RoundTable: Investigating Group Decision-Making Mechanism in Multi-Agent Collaboration"
_ICLR.cc/2025/Conference — Submitted to ICLR 2025_

### Official Review · Reviewer_8kyg · 2024-10-16

**Soundness:** 2
**Presentation:** 3
**Contribution:** 2
**Rating:** 3
**Confidence:** 3

**Summary:**

The paper investigates decentralized decision-making in Multi-Agent Systems (MAS) by introducing a platform called "RoundTable." The authors examine how different social choice mechanisms—like Majority Voting, Unanimous Voting, and Ranked Voting—affect collaboration among agents in a decentralized setting. The paper claims to contribute by analyzing agent conversations to identify linguistic features that signal effective collaboration and proposes several early stopping methods to improve decision efficiency.

However, the paper's contributions are somewhat limited. The social choice mechanisms explored are standard and well-established, raising questions about the novelty of the work. The description of the proposed "RoundTable" platform lacks technical depth, and the experiments, while testing different decision-making methods, do not convincingly demonstrate the practical value or scalability of the approach.

Overall, while the topic is interesting, the paper falls short in terms of innovation and technical rigor.

**Strengths:**

Originality: The paper tackles an important topic in the field of decentralized decision-making within Multi-Agent Systems (MAS). The exploration of social choice methods in this context is relevant, and the effort to introduce linguistic features as a factor in evaluating agent collaboration represents a fresh perspective.

Quality: The paper presents a well-organized experimental setup.

Clarity: The structure of the paper is clear.

Significance:  The paper’s focus on decision-making and communication patterns could lead to practical improvements in the design of MAS systems that operate in complex environments without a central authority.

**Weaknesses:**

1. Lack of Novelty in Decision-Making Mechanisms: The core contribution of the paper revolves around applying well-established social choice mechanisms (e.g., Majority Voting, Ranked Voting, Cumulative Voting) to decentralized Multi-Agent Systems (MAS). However, these voting methods have been extensively studied in both social choice theory and multi-agent systems, which limits the paper’s originality.

2. Limited Theoretical Foundation for Early Stopping Methods: The proposed early stopping methods for agent collaboration, based on linguistic cues and information difference, are interesting but lack rigorous theoretical justification. The idea of using dialogue act transitions and message embedding distances to halt collaboration is novel, but the connection between these metrics and optimal decision-making outcomes is not well substantiated.

3. Scalability and Complexity Issues Not Addressed: The paper does not address key challenges related to the scalability of the RoundTable platform. For example, it is unclear how the proposed platform would handle the computational complexity of aggregating agent votes or resolving conflicts in environments with hundreds of agents. Additionally, while decentralized systems are inherently more scalable than centralized ones, the paper does not explore the trade-offs between decision quality and computational efficiency in larger systems.

4. Other issues:  1)The paper seems to suggest that the group utility is maximized when agents' preferences are aggregated, but the mechanism for achieving this is unclear. 2)How exactly does the context C_i  influence the proposals made by agents? Is the function f_p
  deterministic, or does it incorporate stochastic elements based on agent behavior? Moreover, there is no clear explanation of how the proposals are adjusted or optimized over multiple rounds of collaboration.

**Questions:**

1. Can the authors clarify what specific innovations or adaptations have been made to these traditional mechanisms for use in MAS?
2. Can the authors provide more details about how the context C_i  influences the proposals and voting behaviors of agents? Additionally, are these functions deterministic or stochastic?
3.  The paper does not explicitly address how the social choice function F handles ties or conflicts between agents during voting.

---

> ### Author Response · Authors · 2024-11-19
> **Thank you for your review.**
>
> Thank you for taking the time to review our work and for raising such thoughtful questions and concerns. We have thoroughly evaluated your feedback, addressed the identified weaknesses, and provided detailed responses to resolve your concerns.
>
> **W1: Lack of Novelty in Decision-Making Mechanisms: The core contribution of the paper revolves around applying well-established social choice mechanisms (e.g., Majority Voting, Ranked Voting, Cumulative Voting) to decentralized Multi-Agent Systems (MAS). However, these voting methods have been extensively studied in both social choice theory and multi-agent systems, which limits the paper’s originality.**
>
> We appreciate the reviewer’s observation regarding the use of well-established social choice mechanisms in our work. Below, we clarify the unique contributions and the motivations of our study:
>
>
> 1. **Focus on Behavior Analysis in Decentralized MAS**: While the mechanisms themselves (e.g., Majority Voting, Ranked Voting, Cumulative Voting) are well-established, the novelty of our work lies in analyzing their impact on collaboration dynamics and decision-making behaviors in LLM-based decentralized MAS. This perspective is underexplored in the existing literature, especially in the context of decentralized agent collaboration platforms like RoundTable. Text-based communication has only recently become viable, and the community currently lack the understanding of its limitations and benefits.
> 2. **Motivation for Challenging Default Assumptions**: A key motivation for this work stems from the prevalent use of majority voting as a default in MAS studies. These choices are often made without empirical evidence or scientific scrutiny regarding their effects on collaboration efficiency, fairness, or decision quality. Our study addresses this gap by systematically comparing the impacts of multiple social choice mechanisms.
> 3. **Grounding in Well-Established Definitions for Scientific Rigor**: To draw meaningful and generalizable conclusions, our study deliberately employs established definitions of social choice methods. This approach ensures that our findings are not tied to ad hoc mechanisms but can instead be compared and validated across domains.
> 4. **Key Insights and Contributions**:
>     1. Our findings demonstrate that the choice of social choice mechanism significantly influences collaboration outcomes. For example, score-based mechanisms (e.g., Ranked and Rated Voting) facilitate more nuanced agent preferences, yielding higher performance and fairness compared to one-vote mechanisms (e.g., Majority Voting).
>     2. We show that rigid mechanisms, like Unanimous Voting, reduce collaboration efficiency, while more flexible mechanisms strike a balance between fairness and decision quality.
>     3. Our results also provide actionable insights into optimizing MAS collaboration, such as the benefits of early stopping and combining mechanisms for different stages of collaboration.
>
> We hope this clarification highlights the originality and impact of our work, which lies not in the mechanisms themselves but in the novel analysis of their effects on LLM-driven MAS collaboration.

---

> ### Author Response · Authors · 2024-11-19
>
> **W2: Limited Theoretical Foundation for Early Stopping Methods: The proposed early stopping methods for agent collaboration, based on linguistic cues and information difference, are interesting but lack rigorous theoretical justification. The idea of using dialogue act transitions and message embedding distances to halt collaboration is novel, but the connection between these metrics and optimal decision-making outcomes is not well substantiated.**
>
> We thank the reviewer for recognizing the novelty of our proposed early stopping methods. Below, we address the concerns about their theoretical foundation:
>
>
> 1. **Comprehensive Definitions and Implementation Details**: We have provided a rigorous definition and explanation of each early stopping method in Section 4.5, with detailed algorithms and supporting information included in Appendix C.7. This ensures transparency and replicability, establishing a foundational understanding of how the methods operate and their intended impact on multi-agent collaboration.
> 2. **Empirical Validation Supporting Theoretical Justification**: Our experimental results (Section 5.4, Table 2, and Table 10) demonstrate the effectiveness of these methods in improving decision-making outcomes across different collaboration environments (e.g., Exchange Economy, Recommendation System) and social choice methods (e.g., Majority Voting, Rated Voting). Notably, methods leveraging linguistic cues and information difference consistently outperform baselines and simpler heuristics. This empirical evidence substantiates the connection between these metrics and optimal decision-making outcomes.
> 3. **Connection Between Metrics and Decision-Making Outcomes**:
>     1. **Dialogue Act Transitions**: By analyzing the dynamics of dialogue acts (e.g., Inform, Propose, Evaluate) in collaboration, we identify patterns that indicate progress or stagnation in decision-making. For example, increased transitions involving “Evaluate” or “Accept” acts are often correlated with convergence toward consensus, supporting their use as indicators for halting collaboration at optimal points.
>     2. **Information Difference**: As detailed in Section 5.3.1, a steady decrease in message embedding distances reflects diminishing novelty in agent communication. This metric aligns with the principle that decision-making benefits decrease as discussions become repetitive, justifying its role in terminating collaboration to avoid unnecessary iterations.
> 4. **Broader Implications and Robustness**: While our methods are grounded in observed patterns and empirical performance, their theoretical underpinnings align with principles of efficiency and convergence in multi-agent collaboration. The results in diverse settings and with varying social choice methods suggest that these metrics capture universal aspects of collaboration dynamics.
>
> The current work provides both clear definitions and robust experimental evidence to substantiate the practical utility of our early stopping methods.

---

> ### Author Response · Authors · 2024-11-19
>
> **W3. Scalability and Complexity Issues Not Addressed: The paper does not address key challenges related to the scalability of the RoundTable platform. For example, it is unclear how the proposed platform would handle the computational complexity of aggregating agent votes or resolving conflicts in environments with hundreds of agents. Additionally, while decentralized systems are inherently more scalable than centralized ones, the paper does not explore the trade-offs between decision quality and computational efficiency in larger systems.**
>
> We appreciate the reviewer’s observations regarding scalability and complexity in the RoundTable platform. Below, we provide clarifications and highlight the aspects of scalability addressed in our work:
>
>
> 1. **Scalability Analysis in Ablation Studies:** We explicitly address scalability in Appendix D.1 and Table 6 through ablation studies examining the performance of multi-agent systems under different agent counts. Our results demonstrate that:
>     1. As the number of agents increases, the overall performance generally decreases, indicating a scalability challenge.
>     2. MAS with majority voting exhibits a unique issue with even numbers of participants, where agreement becomes harder to achieve, further highlighting the scalability trade-offs of different social choice mechanisms.
>
>     In the camera ready version, we will move this table to the main text to emphasize the scalability.
>
> 2. **Limitations of Decentralized Systems in Scalability**: We respectfully disagree with the notion that decentralized systems are inherently more scalable than centralized ones in our setting. In fact, we observe:
>     1. **Contextual Complexity**: In decentralized systems, the context size for each agent grows with the number of agents, as all agents must process and respond to a larger amount of information. This stands in contrast to centralized systems, where hierarchical structures can reduce context size by segmenting decisions into smaller, manageable branches.
>     2. **Impact on LLM Outputs**: As noted in our experiments, the increase in agents leads to higher rates of formatting errors in LLM-generated outputs. This highlights the limitations of current LLMs in handling the cognitive load introduced by larger groups in decentralized settings.
> 3. **Trade-offs Between Decision Quality and Computational Efficiency**: While our work primarily focuses on collaboration dynamics and decision quality, our results implicitly explore trade-offs with computational efficiency:
>     1. **Efficiency vs. Quality**: Our early stopping methods (Section 5.4) aim to mitigate inefficiency by halting collaboration when decision quality plateaus, reducing unnecessary computational overhead in larger groups.
>     2. **Social Choice Mechanism Comparison**: We show that simpler mechanisms, such as Majority Voting, may be less computationally intensive but sacrifice nuance and efficiency compared to score-based mechanisms like Rated or Ranked Voting.
>
>
> We hope these clarifications adequately address the reviewer’s concerns and demonstrate our acknowledgment of scalability challenges, along with the steps we have taken to analyze and mitigate them in the context of this study.

---

> ### Author Response · Authors · 2024-11-19
>
> **W4.1: The paper seems to suggest that the group utility is maximized when agents' preferences are aggregated, but the mechanism for achieving this is unclear.**
>
> We appreciate the reviewer’s feedback regarding the mechanism for achieving group utility maximization. Below, we clarify our position and methodology:
>
>
> 1. **Clarification on Group Utility Maximization**: We explicitly state in the paper (Line 212): "...while equilibrium doesn’t guarantee maximum utility, it must lie within one of the equilibria, helping to assess whether conflicts between agents hinder the group’s progress toward the ultimate goal." This highlights that our study does not assume or claim that agents’ preference aggregation guarantees the maximization of group utility.
> 2. **Advantage of the Exchange Economy as an Experimental Environment**: The exchange economy is intentionally chosen as an experimental environment because it reflects the inherent dynamics of MAS, where agents prioritize their individual utilities. The task aims to investigate whether collaborative behaviors—such as negotiation, communication, and compromise—can collectively drive the group toward higher utility outcomes, even as individual agents pursue distinct goals.
> 3. **Exploration of Collaboration Dynamics in a Plus-Sum Game**:
>     1. As noted in Section 4.1, the exchange economy is a plus-sum game where better allocations leading to higher group utility exist before equilibrium is reached. However, achieving these allocations requires agents to engage in behaviors like communication, negotiation, and exploration of trade-offs.
>     2. Our analysis focuses on whether agents’ collaboration behaviors, as influenced by social choice mechanisms and other factors, can overcome conflicts and drive progress toward these optimal allocations.
>
>
> We hope this response clarifies our approach and emphasizes the deliberate focus of the study on analyzing the interplay between individual agent goals and group utility outcomes in a decentralized MAS setting.
>
>
> **W4.2: How exactly does the context C_i influence the proposals made by agents? Is the function f_p deterministic, or does it incorporate stochastic elements based on agent behavior?**
>
> We appreciate the reviewer’s interest in understanding how context C_i influences agent proposals and the nature of the function f_p. Below, we provide clarifications:
>
>
> 1. **Role of Context C_i**: As detailed in Appendix A.2, C_i includes each agent’s unique background, information access, and collaboration history. This context directly informs the agent’s reasoning and proposal generation by shaping its approach to maximizing individual utility.
> 2. **Prompt Design and Implementation**: The design of f_p, the proposal generation function, is guided by the prompt setup described in Appendix A.2. While f_p conceptually operates deterministically, its implementation relies on large language models. Our choice of LLMs (gpt-4o-mini, gpt-3.5-turbo, gpt-4o, llama3.1-8b, llama3.1-70b) inherently introduce variability.
> 3. **Stochasticity of LLM Outputs**: Although we used temperature = 0 (Line 1063) to minimize randomness, LLMs cannot produce fully deterministic outputs due to their underlying probabilistic nature. This stochastic element reflects the flexibility of LLMs to adapt to nuanced inputs, aligning with our goal of simulating realistic agent behavior in collaborative settings.
>
>
> We hope this explanation clarifies the influence of C_i and the deterministic-stochastic interplay in f_p.
>
> **W4.3: Moreover, there is no clear explanation of how the proposals are adjusted or optimized over multiple rounds of collaboration.**
>
> The adjustment of proposals over multiple rounds is facilitated through iterative updates based on shared context and collaboration history (C_i). Each round allows agents to incorporate new information gained from other agents’ messages and decisions, as detailed in Section 3.1 and Appendix A.2. This iterative process enables agents to refine their proposals dynamically, fostering more informed and collaborative decision-making.

---

> > ### Comment · Reviewer_8kyg · 2024-11-22
> >
> > Thank you for your thoughtful response. That said, incorporating additional experiments could make your explanation more compelling.

---

### Official Review · Reviewer_Jxu7 · 2024-10-29

**Soundness:** 2
**Presentation:** 2
**Contribution:** 1
**Rating:** 3
**Confidence:** 4

**Summary:**

This paper looks at how LLM based agents can collaborate via social choice methods.

While this is a natural and potentially interesting area, I found the paper to be weak. The understanding of social choice demonstrated is superficial, and the paper doesn't seem to be aware of the VAST literature on computational social choice. The scientific setup of the experiments seemed somewhat arbitrary.

Overall, I found it hard to understand what the paper was really contributing. It feels more like a workshop paper or a preliminary work than an archival conference paper. I found it very hard to work up any enthusiasm for the work.

**Strengths:**

The idea of looking at social choice mechanisms in the context of LLM agents is natural and compelling.

**Weaknesses:**

The paper doesn't seem to be aware of the literature on computational social choice:

https://cgi.cse.unsw.edu.au/~haziz/comsoc.pdf

Also doesn't seem to be aware of literature on automated negotiation and dialogue:

https://en.wikipedia.org/wiki/Automated_negotiation

I'm puzzled as to what the value of the results is. The setup seems very arbitrary.

**Questions:**

Can you justify your experimental setup?

Are you aware of the literature on computational social choice?

Are you aware of the literature on automated argumentation?

Are you aware of the literature on automated dialogue?

---

> ### Author Response · Authors · 2024-11-19
> **Thank you for your review.**
>
> Thank you for your additional resources to our work. We appreciate the opportunity to clarify our contributions and contextualize our work within the broader literature.
>
> **Rooted in Computational Social Choice and Automated Negotiation/Dialogue**
>
> We are well aware of the fields of computational social choice and automated negotiation and dialogue. In fact, our work builds upon these foundations while addressing a novel context: **LLM-based decentralized multi-agent systems**. While computational social choice has traditionally been applied to static preference aggregation and automated negotiation focuses on task-specific agent behavior, our research advances these areas by:
>
> 1. Analyzing **dynamic, natural-language-based multi-agent** collaboration where agents not only generate and process natural language but also engage in reasoning, negotiation, and decision-making.
> 2. Exploring how various social **choice mechanisms** impact outcomes in environments driven by adaptive and context-aware agents.
>
> Our paper is already grounded in game theory and economics, as reflected in our citations of foundational works such as Arrow (2012) and Black et al. (1958).
>
> **Experimental Setup Justification**
>
> We carefully designed our experimental setup to investigate the impact of social choice mechanisms in LLM-based decentralized multi-agent systems under controlled yet diverse conditions.
>
> 1. **Exchange Economy:** This environment provides a well-defined, quantifiable setting for evaluating collaboration dynamics. Its multiple equilibria and the presence of conflicts make it ideal for testing how different social choice methods influence convergence and fairness.
> 2. **Recommendation System:** This more complex environment introduces strong information asymmetry, requiring agents to collaboratively reason and synthesize knowledge—a scenario well-suited to real-world applications like group decision-making and negotiation.
>
> These setups are not arbitrary; they were chosen to:
>
> 1. Highlight the unique strengths and limitations of various social choice mechanisms.
> 2. Examine the interplay between collaboration type (competitive vs. cooperative) and agent behavior.
> 3. Provide generalizable insights into multi-agent collaboration.
>
> **Value of Results**
>
> Our results demonstrate:
> 1. **Impact of Social Choice**: We show how different mechanisms—e.g., Majority vs. Cumulative Voting—yield distinct collaboration patterns and outcomes, filling a gap in the literature where majority voting is often used by default.
> 2. **Linguistic Insights**: By analyzing dialogue acts and linguistic features, we reveal indicators of effective collaboration, such as information richness and dialogue act transitions. This provides actionable insights for designing more efficient multi-agent systems.
> 3. **Early Stopping Methods**: Our experiments on early stopping methods contribute practical strategies to prevent diminishing returns and optimize decision-making in multi-agent collaboration.
>
> We believe these contributions are valuable to the community and provide intuition for researchers and practitioners designing LLM-based systems.

---

### Official Review · Reviewer_CDcQ · 2024-11-01

**Soundness:** 2
**Presentation:** 3
**Contribution:** 2
**Rating:** 5
**Confidence:** 3

**Summary:**

The paper mainly investigates the phenomenon of group decision-making processes in multi-agent systems (MAS), without focusing on specific algorithm design. Leveraging the RoundTable platform, the paper assesses the efficacy of various social choice mechanisms, such as majority voting and score-based voting, in simulated collaborative scenarios. The core aim of the study is to understand how these decision-making mechanisms affect agent interactions, decision quality, and communication patterns.

**Strengths:**

1. Originality
The paper is original in its approach to studying decentralized decision-making within multi-agent systems (MAS). Unlike many works that focus on designing new algorithms for MAS, this paper shifts focus to understanding the effects of different social choice mechanisms, such as majority voting and score-based voting. This perspective offers fresh insights into the dynamics of agent interactions and decision-making efficiency.
2. Quality
The paper delves into the communication dynamics among agents, early stopping methods, and the relative efficiency of different decision-making processes. This analysis strengthens the study's overall contribution to the field
3. Clarity
The paper clearly outlines its objectives, focusing on how different voting mechanisms influence decision-making in MAS.
4. Significance
The results on early stopping methods and communication analysis could have practical implications, helping to optimize the design of MAS for faster and more effective decision-making processes.

**Weaknesses:**

1.  A primary limitation of the paper is its heavy dependence on simulated environments for assessing the performance of various social choice mechanisms. While simulations offer a controlled context ideal for variable isolation, they often fall short in capturing the intricacies and unpredictability inherent in real-world multi-agent systems (MAS)。
2. The paper introduces an intriguing concept of early stopping methods grounded in the linguistic nuances of agent communication. However, the evaluation of these methods is confined to the specific scenarios outlined in the study, without extensive validation across a broader spectrum of decision-making contexts or diverse agent configurations. This limitation hampers the ability to ascertain the broader efficacy of the early stopping methods in different MAS environments or amidst varied communication dynamics.
3. Although the paper delves into multiple voting mechanisms and scrutinizes their efficacy in multi-agent collaboration, it does not offer a thorough comparison with current state-of-the-art approaches in decentralized decision-making .  The omission of such comparisons restricts the capacity to appraise how the proposed methods measure up against the most effective techniques in the domain.

**Questions:**

1.A deeper analysis of how different voting mechanisms influence group dynamics could add an interesting dimension to the study. If any insights or patterns were observed during the experiments, including them in the paper could provide valuable context for understanding the broader impact of each voting method.
2.Why were comparisons with other state-of-the-art methods in decentralized decision-making not included in the study? The methods currently compared are too ordinary; comparing with some cutting-edge methods might more effectively demonstrate the effectiveness of the proposed approaches.

---

> ### Author Response · Authors · 2024-11-19
> **Thank you for your review.**
>
> Thank you for reviewing our work and raising these insightful questions and concerns. We have carefully considered your feedback, addressed the weaknesses you identified, and resolved your concerns through detailed responses.
>
> **W1: A primary limitation of the paper is its heavy dependence on simulated environments for assessing the performance of various social choice mechanisms. While simulations offer a controlled context ideal for variable isolation, they often fall short in capturing the intricacies and unpredictability inherent in real-world multi-agent systems (MAS)。**
>
>
> We would like to emphasize that our study is not limited to purely simulated environments. Specifically, we conducted experiments in two distinct settings (Section 4, Appendix C.1):
>
> 1. **Exchange Economics (Simulated Environment)**: This environment was chosen for its ability to isolate and analyze specific competitive collaboration behaviors among agents. (Section 4.1, Appendix C.2)
> 2. **Recommendation Systems (Complex Real-World Environment)**: This represents a practical, real-world use case where agents collaborate to achieve a common goal through information-sharing and cooperative decision-making. (Section 4.2, Appendix C.4)
>
> **Rationale for Experimental Environments:**
>
> Our choice of environments was driven by a thoughtful consideration of representativeness. To ensure the validity of our findings, we prioritized selecting tasks that are both meaningful and reflective of scenarios where multiple agents can derive mutual benefits from collaboration. Specifically:
>
>
> 1. **Exchange Economics:** This environment models a competitive, resource-limited scenario. While it is designed as a plus-sum game, agents must navigate the trade-offs of allocating fixed resources, with additional goods providing positive utility. This setup allows us to observe whether agents can engage in effective collaboration despite pursuing individual utility maximization.
> 2. **Recommendation Systems:** In contrast, this environment embodies a cooperative framework where essential information for generating recommendations is distributed among agents. Success in this setting relies on agents’ ability to collaboratively identify information gaps, coordinate actions, and integrate findings to make accurate predictions. This task is inherently real-world in nature, reflecting applications such as collaborative filtering and multi-agent recommendation engines used in practical settings.
>
> **Generalizability and Implications:**
>
> Both environments—competitive (Exchange Economics) and cooperative (Recommendation Systems)—were intentionally selected for their unique characteristics that expose different aspects of collaboration behaviors. By incorporating both:
>
> 1. We ensure that our findings are not limited to a single type of interaction but span a spectrum of collaboration dynamics.
> 2. These environments serve as representative proxies for real-world application scenarios, enabling us to derive meaningful insights about multi-agent collaboration in diverse contexts.
>
> **Early Stage Exploration in Decentralized MAS:**
>
> It is important to note that research in this direction is still in its beginning stages, with limited prior work exploring the dynamics of decentralized multi-agent collaboration in such varied settings. While simulated environments are not a perfect reflection of real-world complexity, their controlled nature allows us to isolate and analyze foundational behaviors systematically. By combining a simulated environment with a complex real-world setting, we leverage the unique characteristics of each, ensuring they are representative and complementary. This dual approach provides a strong starting point for uncovering meaningful insights and advancing research in this emerging field.
>
> **Conclusion:**
>
> In summary, we believe our study’s inclusion of both a simulated yet structured competitive environment (Exchange Economics) and a real-world, complex cooperative environment (Recommendation Systems) robustly reflects the intricacies of agent collaboration. This combination allows us to provide valuable insights into both the theoretical and practical aspects of MAS. We hope this clarification addresses your concern and demonstrates the representativeness and relevance of our experimental design.

---

> ### Author Response · Authors · 2024-11-19
>
> **W2: The paper introduces an intriguing concept of early stopping methods grounded in the linguistic nuances of agent communication. However, the evaluation of these methods is confined to the specific scenarios outlined in the study, without extensive validation across a broader spectrum of decision-making contexts or diverse agent configurations. This limitation hampers the ability to ascertain the broader efficacy of the early stopping methods in different MAS environments or amidst varied communication dynamics.**
>
> We appreciate the observation regarding the evaluation scope of our early stopping methods. However, we would like to clarify that our study encompassed a broad range of decision-making contexts and diverse agent configurations, which substantially validates the generalizability of our approach (Table 10, 11).
>
>
> 1. **Diverse Decision-Making Contexts:** Our experiments spanned two distinct environments, Exchange Economics and Recommendation Systems, which represent competitive and cooperative multi-agent interactions, respectively. These contrasting settings provide a wide spectrum of decision-making scenarios.
> 2. **Variety of Agent Configurations:** To ensure robust validation, we tested our early stopping methods under various collaboration types (competitive and cooperative) and across multiple social choice mechanisms, including unanimous, majority, plurality, rated, ranked, and cumulative voting (Section 4.5, 5.4).
>
> By conducting experiments under such diverse conditions, we demonstrated that our early stopping methods perform consistently and effectively across different agent configurations and communication dynamics. This comprehensive evaluation underscores the adaptability and reliability of our methods in handling both competitive and cooperative multi-agent systems.
>
> **Conclusion:**
>
> We believe that the combination of diverse decision-making contexts and agent configurations provides strong evidence for the broader efficacy of our early stopping methods. This approach ensures that the methods are not confined to a narrow set of scenarios but are instead applicable to a wide range of MAS environments. We hope this addresses your concern and highlights the rigorous evaluation process we employed in our study.

---

> ### Author Response · Authors · 2024-11-19
>
> **W3: Although the paper delves into multiple voting mechanisms and scrutinizes their efficacy in multi-agent collaboration, it does not offer a thorough comparison with current state-of-the-art approaches in decentralized decision-making. The omission of such comparisons restricts the capacity to appraise how the proposed methods measure up against the most effective techniques in the domain.**
>
> **Q2: Why were comparisons with other state-of-the-art methods in decentralized decision-making not included in the study? The methods currently compared are too ordinary; comparing with some cutting-edge methods might more effectively demonstrate the effectiveness of the proposed approaches.**
>
> We appreciate the suggestion to compare our methods with current state-of-the-art approaches in decentralized decision-making. However, the primary focus of our study was to analyze the impact of different social choice mechanisms on collaborative behavior in decentralized MAS, rather than directly benchmarking against existing approaches.
>
> 1. **Unique Focus of Our Study:** Our study aims to understand the broader implications of various social choice methods on collaboration dynamics. This distinctive focus provides novel insights into how different voting mechanisms influence agent collaboration.
> 2. **Specialized Platform for Scientific Comparison:** Our decentralized MAS platform, RoundTable, was specifically designed to enable rigorous scientific comparison of different social choice methods while maintaining well-controlled experimental conditions. Unlike publicly available platforms, RoundTable allows us to:
>     1. Apply a wide range of social choice methods.
>     2. Control external variables that might influence collaboration outcomes.
>
>     These features ensure that our findings are not confounded by extraneous factors, providing a clear understanding of the effects of social choice mechanisms.
>
> 3. **Limitations of Other Platforms:** It is important to note that research in this direction is still in its beginning stages, with limited prior work exploring the dynamics of decentralized multi-agent collaboration in such varied settings. Existing multi-agent platforms often lack the ability to incorporate diverse social choice methods or control critical experimental variables. This makes them unsuitable for the specific goals of our study, which require isolating the effects of different voting mechanisms.
>
> **Conclusion:**
>
> While a direct comparison with other approaches in decentralized decision-making was beyond the scope of this work, our study addresses an important and complementary gap by focusing on the collaborative behaviors elicited by different social choice methods. The unique capabilities of our RoundTable platform allow us to conduct controlled, systematic evaluations that provide valuable insights into agent collaboration. We hope this clarification highlights the distinct contributions of our work and its relevance to the MAS community.

---

> ### Author Response · Authors · 2024-11-19
>
> **Q1: A deeper analysis of how different voting mechanisms influence group dynamics could add an interesting dimension to the study. If any insights or patterns were observed during the experiments, including them in the paper could provide valuable context for understanding the broader impact of each voting method.**
>
> Thank you for the suggestion and interest in further analyzing the influence of different voting mechanisms on group dynamics. We are pleased to share some interesting patterns observed during our experiments, which will also be included in the appendix of the paper:
>
>
> 1. **Strategic Behavior Based on Rounds:** Agents demonstrated awareness of the total number of rounds in the collaboration process. For instance, they often began with greetings and information gathering in the early rounds. As the final round approached, agents became more focused on reaching a consensus, showing increased openness to compromise.
> 2. **Impact of Agent Activity Levels:** Collaboration tended to stall in scenarios where all agents exhibited "lazy" behavior, characterized by a lack of active efforts to find better consensus. However, the presence of even a single "active" agent significantly influenced the group’s dynamics, keeping the collaboration process active and productive.
>
> These observations provide valuable insights into how voting mechanisms and agent behaviors interact to shape group dynamics. We also recognize the potential for further exploration and plan to conduct additional experiments to draw meaningful conclusions for future work. We hope these findings add valuable context to our study.

---

> > ### Comment · Reviewer_CDcQ · 2024-11-25
> > **Thank the authors for the feedback**
> >
> > Some concerns like comparison with approaches in decentralized decision-making have been partly addressed. I have also read other reviewers’ comments, and I would like to keep my score.

---

### Official Review · Reviewer_WAFz · 2024-11-01

**Soundness:** 3
**Presentation:** 4
**Contribution:** 4
**Rating:** 8
**Confidence:** 4

**Summary:**

The authors studies multi-agent collaboration in group decision-making. They investigate the impact of various social choice methods (voting mechanisms) on group decision making. The collaborative platform is evaluated under two environments: simulated exchange economy and a complex recommendation systems both with different metrics to evaluate the performance. Experiments reveal the impact of different social choice methods on the MAS performance. Linguistic features of agent conversations are indicative of effective collaboration that provide useful information for terminating the conversation. The paper provides insights in designing MAS environments using LLMs which can outperform a single agent LLM that is known to have unstable predictions.

**Strengths:**

- The proposed platform, RoundTable, uniquely examines how social choice mechanisms influence collaborative behavior and decision-making in MAS. This offers a novel exploration of decentralized decision-making in MAS

- The findings on linguistic indicators of effective collaboration and the utility of early stopping mechanisms have practical implications for optimizing MAS environments in real-world applications such as recommendation systems and market simulations.

- The paper’s structure is logically organized, with clear explanations of experimental setups (simulated and complex environments) and detailed descriptions of social choice methods and communication metrics. The figures and tables, like those illustrating performance differences across social choice mechanisms, provide visual clarity that aids understanding.

- The authors employ rigorous evaluation metrics (utility, fairness, and efficiency) and analyze cross-agent conversation features to support their conclusions. These metrics give robustness to the study's claims about the advantages of decentralized collaboration and social choice methods.

**Weaknesses:**

- The study uses simulated environments and tasks like the MovieLens recommendation system and an exchange economy setup. Although these are relevant, real-world MAS applications may present further challenges not fully addressed by the study’s current models and assumptions.
-  The paper explores early stopping methods to enhance MAS efficiency, but this approach may not translate as effectively in highly dynamic environments where task complexity varies. More varied scenarios could strengthen the analysis of when and how stopping criteria should adjust to different environments.

- Dependence on LLM-based Agents: The research relies on Large Language Models (LLMs) for MAS agent communication, which may limit the study’s applicability to non-LLM-based MAS systems. This reliance could restrict the generalizability of findings across different types of MAS, especially where language-based communication is less prominent. Otherwise the platform is only specific to LLM-based MAS

**Questions:**

-How does the design of RoundTable compare with existing MAS frameworks in terms of scalability? Would it perform as effectively with a larger number of agents or more complex tasks?
- How consistent are the findings across multiple simulations? Did any particular social choice method show high variance in performance depending on the environment?
- Were there any notable differences in linguistic behavior (e.g., complexity or message length) across environments? If so, what factors influenced these variations?

---

> ### Author Response · Authors · 2024-11-19
> **Thank you for your review.**
>
> Thank you for reviewing our work and providing such insightful questions and concerns. We have thoroughly reflected on your feedback, addressed the weaknesses you highlighted, and provided detailed responses to resolve your concerns.
>
> **W1: The study uses simulated environments and tasks like the MovieLens recommendation system and an exchange economy setup. Although these are relevant, real-world MAS applications may present further challenges not fully addressed by the study’s current models and assumptions.**
>
> **W2: The paper explores early stopping methods to enhance MAS efficiency, but this approach may not translate as effectively in highly dynamic environments where task complexity varies. More varied scenarios could strengthen the analysis of when and how stopping criteria should adjust to different environments.**
>
> We would like to emphasize that our study is not limited to purely simulated environments. Specifically, we conducted experiments in two distinct settings (Section 4, Appendix C.1):
>
> 1. **Exchange Economics (Simulated Environment)**: This environment was chosen for its ability to isolate and analyze specific competitive collaboration behaviors among agents. (Section 4.1, Appendix C.2)
> 2. **Recommendation Systems (Complex Real-World Environment)**: This represents a practical, real-world use case where agents collaborate to achieve a common goal through information-sharing and cooperative decision-making. (Section 4.2, Appendix C.4)
>
> **Rationale for Experimental Environments:**
>
> Our choice of environments was driven by a thoughtful consideration of representativeness. To ensure the validity of our findings, we prioritized selecting tasks that are both meaningful and reflective of scenarios where multiple agents can derive mutual benefits from collaboration. Specifically:
>
> 1. **Exchange Economics**: This environment models a competitive, resource-limited scenario. While it is designed as a plus-sum game, agents must navigate the trade-offs of allocating fixed resources, with additional goods providing positive utility. This setup allows us to observe whether agents can engage in effective collaboration despite pursuing individual utility maximization.
> 2. **Recommendation Systems**: In contrast, this environment embodies a cooperative framework where essential information for generating recommendations is distributed among agents. Success in this setting relies on agents’ ability to collaboratively identify information gaps, coordinate actions, and integrate findings to make accurate predictions. This task is inherently real-world in nature, reflecting applications such as collaborative filtering and multi-agent recommendation engines used in practical settings.
>
> **Generalizability and Implications**:
>
> Both environments—competitive (Exchange Economics) and cooperative (Recommendation Systems)—were intentionally selected for their unique characteristics that expose different aspects of collaboration behaviors. By incorporating both:
>
>
> 1. We ensure that our findings are not limited to a single type of interaction but span a spectrum of collaboration dynamics.
> 2. These environments serve as representative proxies for real-world application scenarios, enabling us to derive meaningful insights about multi-agent collaboration in diverse contexts.
>
> **Early Stage Exploration in Decentralized MAS**
>
> It is important to note that research in this direction is still in its beginning stages, with limited prior work exploring the dynamics of decentralized multi-agent collaboration in such varied settings. While simulated environments are not a perfect reflection of real-world complexity, their controlled nature allows us to isolate and analyze foundational behaviors systematically. By combining a simulated environment with a complex real-world setting, we leverage the unique characteristics of each, ensuring they are representative and complementary. This dual approach provides a strong starting point for uncovering meaningful insights and advancing research in this emerging field.
>
> **Conclusion:**
>
> In summary, we believe our study’s inclusion of both a simulated yet structured competitive environment (Exchange Economics) and a real-world, complex cooperative environment (Recommendation Systems) robustly reflects the intricacies of agent collaboration. This combination allows us to provide valuable insights into both the theoretical and practical aspects of MAS. We hope this clarification addresses your concern and demonstrates the representativeness and relevance of our experimental design.

---

> ### Author Response · Authors · 2024-11-19
>
> **W3: Dependence on LLM-based Agents: The research relies on Large Language Models (LLMs) for MAS agent communication, which may limit the study’s applicability to non-LLM-based MAS systems. This reliance could restrict the generalizability of findings across different types of MAS, especially where language-based communication is less prominent. Otherwise the platform is only specific to LLM-based MAS.**
>
> Our work specifically focuses on LLM-based MAS, distinguishing it from prior research in computational social choice and game theory, which often do not center on language-driven agent interactions. This emphasis enables us to explore unique challenges and opportunities inherent in language-based communication and decision-making mechanisms, setting a distinct scope for our contributions.
>
> **Q1: How does the design of RoundTable compare with existing MAS frameworks in terms of scalability? Would it perform as effectively with a larger number of agents or more complex tasks?**
> Decentralized systems are inherently less scalable than centralized ones. We observe:
>
> 1. **Contextual Complexity**: In decentralized systems, the context size for each agent grows with the number of agents, as all agents must process and respond to a larger amount of information. This stands in contrast to centralized systems, where hierarchical structures can reduce context size by segmenting decisions into smaller, manageable branches.
> 2. **Impact on LLM Outputs**: As noted in our experiments, the increase in agents leads to higher rates of formatting errors in LLM-generated outputs. This highlights the limitations of current LLMs in handling the cognitive load introduced by larger groups in decentralized settings.
>
> However, decentralized MAS is better in the following scenarios:
>
> 1. **Flexible adaptation to complex/variant tasks**. Unrestricted communication between agents enables wide-range of exploration in possible solutions, make the system less over-fitted to a single task by design.
> 2. **Exists information asymmetry**. When each agent is required to make decision independently without sharing full information to others (e.g. poker game, werewolf game), decentralized setting is required.
>
>
> **Q2: How consistent are the findings across multiple simulations? Did any particular social choice method show high variance in performance depending on the environment?**
>
> While other social choices have shown stable outcomes, unanimous voting made collaboration unstable due to its strict acceptance criteria (Table 1). High disagreement ratio under unanimous voting slowed down reaching consensus across agents (Figure 2a). As a trade off, unanimous voting gave the highest fairness in collaboration.
>
>
> **Q3: Were there any notable differences in linguistic behavior (e.g., complexity or message length) across environments? If so, what factors influenced these variations?**
>
> Linguistic behavior well represented the unique characteristics in our environments. We want to highlight two differences across them:
>
> 1. **Message Complexity** (Table 8). In the exchange economy, message complexity peaked in the second round and declined thereafter, reflecting competitive collaboration where agents quickly proposed solutions, aligning with Figure 2a's utility gains, which plateaued after round two. Conversely, the recommendation system showed increasing complexity across rounds as agents continuously explored information, leading to progressively detailed conclusions.
> 2. **Transition between Dialogue Acts** (Figure 4): In the exchange economy, repeated proposals and requests formed the foundation of competitive collaboration among agents, reflecting the behavior of asking a new allocation and other’s preferences. In contrast, the recommendation system focused on requesting new information, highlighting the need for additional data to develop more detailed conclusions.

---

### Author Response · Authors · 2024-11-22
**Appreciation for Reviews and Request for Feedbacks**

Dear Reviewers,

We sincerely appreciate your efforts in reviewing our work and providing valuable feedback. We have carefully addressed your comments in our rebuttal and would be grateful if you could review our responses. Please feel free to let us know if further clarification is needed.

Thank you for your time and thoughtful consideration.

Sincerely,
The Authors

---

### Meta-Review · Area_Chair_4iZ2 · 2024-12-17

**Metareview:**

Reviewers liked the topic of evaluating social choice mechanisms for multiagent group decision making, yet many of them were not convinced by the proposed methodology due to somewhat arbitrary design choices, missing literature reviews and comparisons, unclear messages, and lack of analysis on other important issues such as stopping conditions, scalability, and complexity.

**Additional Comments On Reviewer Discussion:**

The most positive reviewer (WAFz) did not participate in the discussion, so we didn't get a chance to have his/her input. My personal perception is that the paper is not rigorous enough and does not meet the bar of ICLR.

---

### Decision · Program_Chairs · 2025-01-22

Reject